# Top-H Decoding: Adapting the Creativity and Coherence with Bounded Entropy in Text Generation

**Erfan Baghaei Potraghloo$^{u}$†, Seyedarmin Azizi$^{u}$†, Souvik Kundu$^{i}$, and Massoud Pedram$^{u}$**

$^{u}$University of Southern California, Los Angeles, USA
$^{i}$Intel AI, USA
† Equal contribution authors
{baghaeip, seyedarm, pedram}@usc.edu, souvikk.kundu@intel.com

## Abstract

Large language models (LLMs), despite their impressive performance across a wide range of tasks, often struggle to balance two competing objectives in open-ended text generation: fostering diversity and creativity while preserving logical coherence. Existing truncated sampling techniques, including temperature scaling, top-$p$ (nucleus) sampling, and min-$p$ sampling, aim to manage this trade-off. However, they exhibit limitations, particularly in the effective incorporation of the confidence of the model into the corresponding sampling strategy. For example, min-$p$ sampling relies on a single top token as a heuristic for confidence, eventually underutilizing the information of the probability distribution. To effectively incorporate the model confidence, this paper presents ***top-H*** decoding. We first establish the theoretical foundation of the interplay between creativity and coherence in truncated sampling by formulating an **entropy-constrained minimum divergence** problem. We then prove this minimization problem to be equivalent to an **entropy-constrained mass maximization** (ECMM) problem, which is NP-hard. Finally, we present top-H decoding, a computationally efficient greedy algorithm to solve the ECMM problem. Extensive empirical evaluations demonstrate that top-H outperforms the state-of-the-art (SoTA) alternative of min-$p$ sampling by up to **25.63**% on creative writing benchmarks, while maintaining robustness on question-answering datasets such as GPQA, GSM8K, and MT-Bench. Additionally, an *LLM-as-judge* evaluation confirms that top-H indeed produces coherent outputs even at higher temperatures, where creativity is especially critical. In summary, top-H advances SoTA in open-ended text generation and can be *easily integrated* into creative writing applications. The code is available at https://github.com/ErfanBaghaei/Top-H-Decoding.

## 1 Introduction

Large language models (LLMs) have exhibited impressive abilities in open-ended generation tasks, including creative writing and multi-turn dialogue (Lee et al., 2022). However, these models often need to deal with the challenge of *balancing creativity and coherence*, accepting less likely and more imaginative token choices while avoiding incoherent or nonsensical output. This trade-off is complex, as indiscriminate broadening of the sampling pool can lead to fragmented or disjoint text (Holtzman et al., 2019).

To navigate this balance, various sampling strategies have emerged, including temperature scaling (Ackley et al., 1985), top-$k$ (Fan et al., 2018), top-$p$ (nucleus) (Holtzman et al., 2019), $\eta$ (Hewitt et al., 2022), and min-$p$ sampling (Nguyen et al., 2024). They generally apply heuristics to control diversity and risk. Specifically, min-$p$ sampling (Nguyen et al., 2024) stands out for its dynamic

39th Conference on Neural Information Processing Systems (NeurIPS 2025).

truncation of low-probability tokens using a threshold tied to the probability of the top token. Although this method performs well at high temperatures ($T$), its exclusive reliance on the maximum probability token to estimate confidence disregards the potential distribution of the probability mass over the remaining vocabulary. As a result, min-$p$ remains vulnerable to *over-truncation* in sparse (low-entropy) distributions and *under-truncation* in dense (high-entropy) distributions.

The above limitation motivates the need for a more methodical *confidence-aware* sampling framework that accounts for the overall shape of the distribution, rather than only its peak. In addition, the proliferation of heuristic methods highlights a deeper issue, namely *the lack of a theory-based foundation to analyze the interplay between creativity and coherence in autoregressive generation.*

**Our Contributions.** Towards effective incorporation of the confidence of the model, in this work, we present **top-H** decoding. In particular, top-H maintains the creativity and coherence balance guided by bounded entropy in text generation. Unlike most earlier approaches that rely on a fixed threshold, top-H dynamically selects a subset of tokens such that the resulting truncated distribution over the selected subset has an upper-bounded uncertainty while maintaining minimal divergence from the original distribution predicted by the model.

To formally ground top-H, we first introduce a constrained optimization problem that characterizes the trade-off between creativity and coherence in language generation, namely, ***entropy-constrained minimum divergence*** (ECMD). We show that this minimization is equivalent to an ***entropy-constrained mass maximization*** (ECMM) problem. We then prove that ECMM is NP-hard. Thus, in top-H, we offer a greedy solution that is both efficient and practically effective in approximating the solution of the ECMM while bounding the entropy of the selected distribution. During autoregressive generation, as the token distribution evolves at each step, top-H adjusts its entropy threshold based on the entropy of the token distribution, thereby ***dynamically*** adapting to the model's varying confidence over time.

We validate the effectiveness of top-H through extensive experiments in a diverse set of tasks, including creative writing (Alpaca-Eval (Li et al., 2023) and MT-Bench (Zheng et al., 2023)), reasoning (GSM8k (Cobbe et al., 2021) and GPQA (Rein et al., 2024)), and human-aligned evaluations using LLM as a judge framework. Specifically, top-H consistently outperforms existing sampling methods in accuracy while maintaining a robust balance between expressiveness and fluency. For example, compared to min-$p$, top-H demonstrates an accuracy improvement of up to **25.63%**.

## 2 Related Work

### 2.1 Stochastic Sampling Strategies for Autoregressive Models

Temperature scaling (Ackley et al., 1985) multiplies the logits by a scalar, encouraging the exploration of less likely tokens. However, it can get too indiscriminate at high $T$s, generating incoherent or contradictory texts. Top-$k$ (Fan et al., 2018) includes only the $k$ highest probability tokens. Although simple, this *hard cutoff* is insensitive to context, sometimes excluding large swaths of moderately plausible tokens. Top-$p$ (nucleus) sampling (Holtzman et al., 2019) chooses the smallest subset of tokens whose cumulative probability exceeds $p$. This alleviates some of the rigidity of top-$k$. Unfortunately, at high $T$, the distribution can be so flat that the top-$p$ may inadvertently include very low-probability tokens, harming coherence. This incoherence in top-$p$ sampling is demonstrated in the experimental results Table 4, where the coherence score on text drops significantly at higher $T$.

Min-$p$ (Nguyen et al., 2024) sampling dynamically scales a base probability score threshold $p_{base}$ by the probability of the top-1 token. This effectively restricts the sample space more aggressively when the model is confident. Min-$p$ has been shown to outperform top-$p$ in tasks requiring both diversity and correctness at higher temperatures. However, *its reliance on only the highest-probability token* can overlook broader features of the distribution. Two different probability mass functions might share a top-1 token probability; however, they differ widely in their overall confidence.

### 2.2 Entropy-Based Sampling Strategies for Autoregressive Models

Several methods attempt to exploit *entropy* or related uncertainty measures when sampling. $\eta$-sampling (Hewitt et al., 2022) dynamically adjusts the sampling threshold based on the entropy of the distribution of the next token. However, this method often requires carefully tuned hyperparameters and can introduce significant runtime overhead at higher $T$s. Mirostat (Basu et al., 2020) aims to

maintain a target perplexity (related to entropy) via feedback control. Although it can yield steady perplexities, it adds complexity to parameter tuning and integration into generation pipelines.

Despite their *entropy-aware* intentions, these approaches do not strictly limit the randomness of the sampling distribution; instead, they often aim to achieve a perplexity target or modify the sampling heuristics in real-time. As a result, controlling the maximum allowed randomness in the final distribution, thus ensuring both coherence and flexibility, can be challenging.

# 3   Motivational Case Study

This section presents a key motivation to develop a new sampling method, despite the widespread use of nucleus and min-$p$ sampling within the community. Specifically, we try to pose the following question.

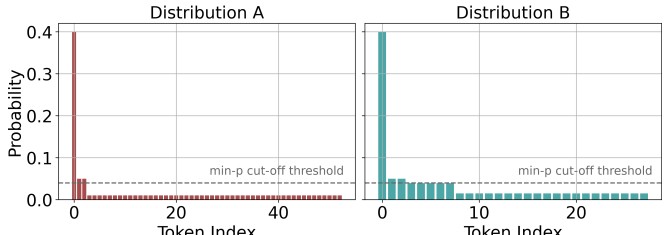

*Why do we need a more distribution-aware sampling technique if min-$p$ already considers the model's confidence?*

Figure 1: Probability distribution of two different types with associated min-$p$ threshold.

Min-$p$ employs a dynamic truncation threshold by modulating the maximum probability of the next token probability distribution with a base factor. Although this approach accounts for the confidence of the model to some extent, it is insufficient to select an optimal sampling pool.

Consider one scenario where min-$p$ may yield low efficacy as illustrated in Fig. 1. Distributions A and B represent two token probability distributions over the vocabulary, where tokens are sorted by their probability values, and tokens not shown have a probability of zero. Since both distributions have the same maximum probability, min-$p$ applies a similar cut-off threshold. However, the two distributions are distinct in terms of confidence. Distribution A exhibits greater randomness, as it contains numerous low-probability tokens, while distribution B includes some high-probability tokens discarded due to min-$p$'s truncation threshold. This example demonstrates that the min-$p$ approach does not accurately capture the underlying distributional characteristics. Consequently, we are motivated to adopt a sampling method that considers the overall shape of the probability distribution rather than solely relying on a maximum probability threshold. In Appendix C.4, we demonstrate how our proposed sampling strategy, top-H, addresses this issue using the exact same example.

# 4   Theoretical Foundation for Entropy-Based Truncation Sampling

This section establishes the theoretical foundations of top-H sampling. Given a language model $\mathcal{M}$ and a preceding context window $x_{1:t-1}$, the probability distribution over the vocabulary $\mathcal{V}$ for the next token $x_t$ can be written as,

$$p(x_t) = \mathcal{M}(x_{1:t-1}). \tag{1}$$

Our objective is to determine a subset $S \subseteq \mathcal{V}$ from which the next token will be sampled, ensuring that the resulting probability distribution over the subset S, denoted $q(x_t) : S \to [0, 1]$ with $\sum_{x_t \in S} q(x_t) = 1$, satisfies the following desired characteristics:

1. **Minimum divergence from the original probability mass function:** The subset $S$ should be constructed such that the distribution over the tokens in the subset, $q(x_t)$, has minimal divergence from the original distribution $p(x_t)$, thereby "maximally matches" $p(x_t)$.

2. **Reduced randomness for enhanced coherence**: The probability mass function $q(x_t)$ should exhibit lower randomness compared to $p(x_t)$ in the sense that $H(q) \leq H(p)$ where $H(.)$ denotes the Shannon entropy, effectively upper-bounding uncertainty.

These criteria form the basis of top-H, which seeks to construct $S$ and calculate $q$ so that $q$ maximally matches $p$ while exhibiting lower randomness compared to it[1]. To regulate diversity in a controllable manner, we introduce a parametric randomness bound, parameterized by $\alpha$ (see Eq. 2). We formalize this objective as a minimization of the Jensen–Shannon divergence (JSD) between $p$ and $q$ under the parametric entropy constraint. Formally, we intend to solve the following.

$$\min_{S} \ \mathrm{JSD}(q \,\|\, p) \quad \text{subject to} \quad H(q) \ \leq \ \alpha \, H(p), \tag{2}$$

where $\alpha \in (0, 1)$ is a tunable hyperparameter. We refer to this problem as ***entropy-constrained minimum divergence*** (ECMD). By upper-bounding $H(q)$ in proportion to $H(p)$, ECMD encourages the sampling of *more* tokens in the case of higher uncertainty (higher $H(p)$) and the *less* token in the case of lower uncertainty (lower $H(p)$). This approach preserves coherent tokens in contexts where the model "knows" likely the next token, yet encourages exploration when multiple candidates plausibly fit the context, precisely where creativity is more beneficial. Therefore, with an appropriate choice of $\alpha$, solving the ECMD problem can *ideally balance creativity and coherence* in autoregressive text generation. In the rest of this section, we prove the following statements. I) Minimizing JSD under an entropy bound is equivalent to maximizing the sum of probabilities of the tokens in $S$ (subject to $H(q) \le \alpha H(p)$). II) The ECMD problem is, in general, NP-hard.

### 4.1 Formulation of the JSD Minimization Problem

We first start by defining the values of each element in the probability distribution of $p$ and $q$, respectively. Assuming $v_i$ denotes the $i^{th}$ token in the dictionary $\mathcal{V}$, the conditional probability $p_i$ of selecting $v_i$ as the $t^{th}$ generated token given $x_{1:t-1}$ is,

$$p_i \ = \ \mathtt{Prob}(x_t = v_i | x_{1:t-1}) \quad \text{for} \quad i = 1, 2 \ldots, n$$

where $n = |\mathcal{V}|$, $|.|$ identifies the cardinality of a set. Similarly, the conditional probability $q_i$ of selecting $v_i$ as the $t^{th}$ generated token given $x_{1:t-1}$ is

$$q_i = \begin{cases} \frac{p_i}{\Gamma_S} & v_i \in S \\ 0 & \text{otherwise.} \end{cases} , \quad \text{where} \quad \Gamma_S = \sum_i p_i \mathbb{1}_{\{v_i \in S\}} \tag{3}$$

Having defined the distributions, the Jensen-Shannon divergence between $p$ and $q$ is calculated as

$$\mathrm{JSD}(p\|q) = \frac{1}{2} D_{KL}(p\|M) + \frac{1}{2} D_{KL}(q\|M), \quad \text{where} \quad M = \frac{1}{2}(p + q) \tag{4}$$

Next, without loss of generality, we use the properties of JSD and re-formulate the ECMD problem as a maximization problem of the probability mass function for ease of analysis.

### 4.2 Equivalence to Entropy-Constrained Mass Maximization

The ECMD problem in Equation 2 is challenging to analyze directly due to the complexity associated with the expansion of JSD. We thus reformulate the problem using the $\Gamma_S$ metric to facilitate analysis and interpretation. The following theorem formalizes the necessary condition for achieving the optimal solution to the original optimization problem in terms of $\Gamma_S$.

**Theorem 1.** *The Jensen-Shannon divergence between the distributions $p$ and $q$ is only dependent on the $\Gamma_S$ and can be minimized by maximizing $\Gamma_S$.*

*Proof.* Refer to Appendix A.1 for the proof. □

As a result, ECMD can be rewritten as the following,

$$\max_{S} \ \Gamma_S \quad s.t. \quad H(q) \le \alpha H(p) \ \rightarrow \ \max_{S} \sum_i p_i \, \mathbb{1}_{\{v_i \in S\}} \quad s.t. \quad H(q) \le \alpha H(p) \tag{5}$$

---

[1]From now on, we will use $p$ and $q$ to refer the distributions of the next token defined over the original set ($\mathcal{V}$) and the selected subset ($S$) of tokens.

We name the above formulation as ***entropy-constrained mass maximization*** (ECMM). This refor­mulated version of the problem is easier to reason about. Next, we prove that given $0 < \alpha < 1$, the problem remains NP-hard. Finally, we propose a greedy approach as a solution to this. Unless otherwise specified, we empirically set $\alpha = 0.4$ and use it throughout our analysis.

### 4.3 NP-Hardness Proof of the ECMM Problem

**Theorem 2.** *The entropy-constrained mass maximization problem is NP-hard.*

*Proof.* In Appendix A.2, we present a detailed polynomial-time reduction from the well-known cardinality-constrained subset-sum (CCSS) problem (Garey & Johnson, 1979). As CCSS is a popular NP-complete problem, our formulation establishes the NP-hardness of ECMM. $\square$

## 5 Top-H Decoding Method

Having established the NP-hardness of the ECMM problem, we recognize that it cannot be solved efficiently in the general cases. Thus, to produce a practical, efficient, and yet competitive solution, we now present a greedy approximation algorithm, namely **top-H**. Top-H incrementally maximizes the objective of Eq. 5, while adhering to the imposed entropy constraint.

---

**Algorithm 1** Top-H: proposed greedy token selection algorithm

---

**Require:** Probability mass function $p = (p_1, p_2, \ldots, p_n)$, entropy threshold coefficient $\alpha \in (0, 1)$
**Ensure:** Selected token set $S$
 1: Sort tokens in descending order of probability: $p_1 \geq p_2 \geq \ldots \geq p_n$
 2: Initialize $S \leftarrow \emptyset$, $H(q) \leftarrow 0$
 3: **for** each token $i$ in sorted order **do**
 4:     Add token $i$ to $S$
 5:     Compute updated distribution $q$ over $S$
 6:     Compute entropy $H(q)$
 7:     **if** $H(q) > \alpha \cdot H(p)$ **then**
 8:         Remove token $i$ from $S$
 9:         **break**
10:     **end if**
11: **end for**
12: **return** $S$

---

Algorithm 1 outlines the token selection strategy of top-H. The objective is to maximize the probability mass of the tokens $\sum_i p_i \, \mathbb{1}_{\{v_i \in S\}}$, where the tokens $v_i$ are selected into the sampling set $S$. To achieve this, the algorithm begins by sorting all candidate tokens in the *descending* order of their probabilities. It then iteratively adds tokens to the sampling set in this order. After each addition, a distribution $q$ is constructed over the selected tokens, and its entropy is calculated. Top-H continues this process until the entropy of $q$ reaches the dynamic[2] threshold $\alpha \cdot H(p)$, ensuring that the selected subset respects the global entropy constraint.

Unlike prior truncation-based sampling methods, top-H explicitly controls the randomness of the distribution it samples from, $H(q)$, by adapting it to the entropy of the original next token probability distribution $H(p)$. As a result, the allowed randomness dynamically adjusts throughout the steps of autoregressive generation as $p$ evolves. In Section 6.2, we provide empirical evidence on the competitiveness of the top-H's greedy approach in solving the ECMM.

We now present a theorem that guarantees the termination of the algorithm, with an *early* convergence governed by the entropy scaling coefficient $\alpha$.

**Termination Guarantee.** Entropy is a *non-linear* and *non-monotonic* function. Thus, the entropy of the distribution $q$ over a set $S$ is not predictable as tokens are added. Specifically, adding a token

---

[2]At each step of auto-regressive token generation, the model produces a new probability distribution $p$, causing the entropy threshold $\alpha H(p)$ to vary dynamically across generation steps.

to $S$ can increase or decrease the entropy, depending on the underlying probabilities. However, under a greedy selection strategy, it can be shown that each additional token strictly increases the entropy of $q$. Consequently, the entropy constraint is not a vacuous bound, and the growth of $S$ is inherently bounded; the set cannot expand indefinitely without eventually violating the entropy constraint. This intuition is formalized in the following theorem.

**Theorem 3.** *Consider a greedy algorithm that selects tokens in descending order of their probabilities. Let $q$ be the probability mass function over the selected tokens. Then, the entropy of $q$ increases strictly at each selection step and is maximized only when all tokens are selected. Therefore, if the entropy threshold coefficient $\alpha$ is chosen such that $0 < \alpha < 1$, the algorithm is guaranteed to terminate before all tokens are selected.*

*Proof.* Refer to Appendix A.3 for the proof. □

The termination guarantee uses the monotonic growth of entropy under the greedy selection procedure. Each token added to the set contributes positively to the entropy, regardless of its probability, thus ensuring that the entropy $H(q)$ approaches the threshold $\alpha H(p)$. The algorithm stops adding tokens to the set $S$ at the moment when any further addition of tokens would violate the constraint. This ensures that the ECMM objective avoids the trivial solution of selecting all tokens while still satisfying the entropy constraint.

## 6 Experiments

### 6.1 Experimental Setup

**Models, sampling methods, and datasets.** We evaluate top-H on three recent instruction-tuned language models, namely, LLaMA3.1–8B–Instruct (Grattafiori et al., 2024), Qwen2.5–3B (Yang et al., 2024), and Phi-3-Mini-4K-Instruct (Abdin et al., 2024). As baselines, we compare with several widely used truncation-based sampling methods, namely, top-$k$, top-$p$ (nucleus sampling), min-$p$, and $\eta$-sampling. Our evaluations span multiple benchmarks designed to test creative generation, reasoning ability, and evaluative judgment. Specifically, we also used the Alpaca-Eval dataset (Li et al., 2023), GSM8K (Cobbe et al., 2021), GPQA (Rein et al., 2024), MT-Bench (Zheng et al., 2023), and an LLM-as-a-judge evaluation setting.

**Experimental settings.** For decoding hyperparameters, we follow the configuration in (Nguyen et al., 2024), using `min_p = 0.1`, `top_p = 0.9`, and $\eta = 0.0002$ for min-$p$, top-$p$, and $\eta$-sampling methods, respectively. We choose the best result out of the k = 10, 20, and 50 for top-$k$ method. Regarding the evaluation, we use the `lm-eval-harness` framework (EleutherAI, 2023) and report exact match accuracy with the flexible extract filter on the GPQA and GSM8K datasets, length-controlled win rate on Alpaca-Eval, and judge scores (on a scale from 1 to 10) on MT-Bench. For Alpaca-Eval and MT-Bench, we used GPT-4o (OpenAI, 2024) as the judge LLM. All experiments were conducted on a single NVIDIA A6000 GPU, and algorithms were implemented using PyTorch version `2.5.1+cu124` and the Hugging Face Transformers library version `4.50.1`.

#### 6.1.1 Performance on Creative Writing: Alpaca-Eval and MT-Bench

Fig. 2 presents compelling evidence for the superiority of top-H sampling compared to alternative SoTA approaches. In Fig. 2(a-c) (Alpaca-Eval), top-H shows remarkable improvements over the state-of-the-art min-$p$ method, and also conventional sampling methods. For example, for LLaMA3.1-8B across different $T$, top-H demonstrates an win-rate (%) improvement of up to $17.11\%$ compared to SoTA min-$p$ sampling. A critical finding from Fig. 2 is the resilience of top-H to temperature scaling. **While traditional sampling methods exhibit severe performance degradation at higher $T$, top-H preserves much of its effectiveness**. For instance, for LLaMA3.1-8B-Instruct in Fig. 2(a), top-$p$ sampling shows a catastrophic $34.06\%$ decline in win rate from $T$=1 to $T$=2. In contrast, top-H experiences only a $3.78\%$ reduction over the same temperature range. This robustness is particularly significant given that higher $T$ settings are essential for generating diverse, creative texts. The MT-Bench results (Fig. 2(d-f)) further validate the capability of top-H. For example, for LLaMA3.1-8B, similar to that on Alpaca-Eval, the advantage becomes more pronounced at higher $T$, with top-H achieving a higher score value of up to $3.78$.

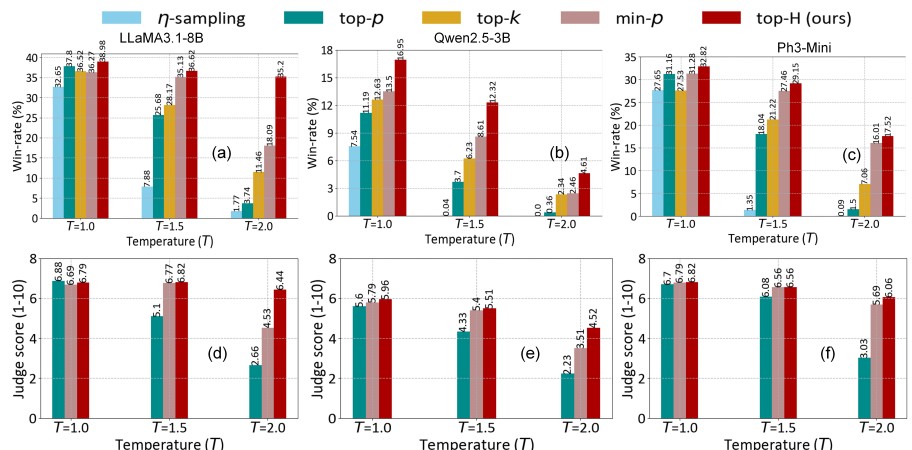

Figure 2: (a)-(c): Length-controlled win rates (%) comparison of different SoTA sampling with top-H on Alpaca-Eval benchmark. (d)-(f): Judge scores (on a scale of 1 to 10) on MT-Bench.

### 6.1.2 Performance on Reasoning and CoT Tasks

Following the setup in (Nguyen et al., 2024) we use the `gsm_cot` (8-shot) and `gpqa_main_generative_n_shot` (8-shot) tasks for GSM8k and GPQA, respectively.

The experimental results in Tables 1 and 2 demonstrate the effectiveness of top-H compared to min-$p$ and top-$p$ sampling across language models and temperature settings. Temperature is a key hyperparameter in generation, striking a balance between creativity and factual accuracy.

On the GSM8k benchmark (Table 1), top-H consistently outperforms the alternatives. At $T$=1, it leads for all the models, outperforming both min-$p$ and top-$p$ by a significant margin. Top-H maintains strong accuracy as temperature increases, while the baselines degrade significantly. At $T$=2, the contrast becomes even more pronounced, with top-$p$ showing near-total collapse (declining by up to 73.62% (Phi-3-Mini) in accuracy), while top-H experiences a very modest degradation, showing an accuracy improvement of up to **25.63**% compared to min-$p$ (on LLaMA3.1-8B).

A similar trend is observed in GPQA benchmark (Table 2). At $T$=1, top-H remains competitive, outperforming both top-$p$ and min-$p$ on Qwen2.5 and Phi-3-Mini. At $T$=2, it exhibits notable robustness, maintaining performance levels substantially higher than those of top-$p$, which experiences significant deterioration. Compared to min-$p$, top-H an accuracy improvement of up to 3.12%, 2.67%, and 7.36% on Qwen2.5, LLaMA3.1, and Phi-3-Mini, respectively. In summary, **top-H demonstrates competitive performance even at low temperatures and significantly superior performance at higher temperatures, marking it as a reliable sampling strategy for diverse generation needs**.

| Temperature | Qwen2.5 3B | | | LLaMA3.1-8B-Instruct | | | Phi-3-Mini | | |
|---|---|---|---|---|---|---|---|---|---|
| | Min-$p$ | Top-$p$ | Top-H | Min-$p$ | Top-$p$ | Top-H | Min-$p$ | Top-$p$ | Top-H |
| 1.0 | 72.40 | 71.27 | **75.97** | 48.90 | 67.93 | **76.35** | 81.96 | 81.35 | **83.24** |
| 1.5 | 66.79 | 55.57 | **72.55** | 58.00 | 23.81 | **70.51** | 77.10 | 67.25 | **77.86** |
| 2.0 | 49.43 | 9.10 | **55.57** | 13.72 | 2.65 | **39.35** | **60.88** | 7.73 | 60.20 |

Table 1: Accuracy (%) for top-H, min-$p$, and top-$p$ on GSM8K.

| Temperature | Qwen2.5 3B | | | LLaMA3.1-8B-Instruct | | | Phi-3-Mini | | |
|---|---|---|---|---|---|---|---|---|---|
| | Min-$p$ | Top-$p$ | Top-H | Min-$p$ | Top-$p$ | Top-H | Min-$p$ | Top-$p$ | Top-H |
| 1.0 | 28.35 | 27.68 | **28.79** | 26.34 | **32.81** | 29.24 | 31.92 | 30.58 | **32.37** |
| 1.5 | **30.13** | 27.23 | 27.90 | 28.35 | 28.57 | **30.58** | 29.91 | 28.57 | **30.80** |
| 2.0 | 25.00 | 22.32 | **28.12** | 26.12 | 23.88 | **28.79** | 23.44 | 18.53 | **30.80** |

Table 2: Accuracy (%) for top-H, min-$p$, and top-$p$ on GPQA.

### 6.1.3 Performance Analysis with LLM-as-a-Judge

In this section, we employ the LLM-as-a-Judge framework to directly evaluate the creativity and coherence of texts generated using min-$p$, top-$p$, and top-H sampling strategies. Following the

evaluation setup proposed in (Nguyen et al., 2024), we use three open-ended prompts designed to elicit creative storytelling on diverse topics. We generate responses using three different models: LLaMA3.1-8B-Instruct, Qwen2.5-3B, and Phi3-Mini-4k-Instruct, each evaluated across three different temperature settings. The top-$p$ and min-$p$ sampling methods serve as baselines for comparison. The prompts used are closely aligned with those in (Nguyen et al., 2024) and are listed in Appendix B. We use GPT-4o (OpenAI, 2024) as the judge model to assess the outputs, which scores the responses based on creativity and coherence using the evaluation prompt detailed in Appendix B.

For each evaluation, the outputs of the three sampling strategies are randomly shuffled to mitigate positional bias. The scores are then extracted from the GPT-4o evaluation responses. To reduce the impact of randomness and noise, each experimental configuration, defined by model, temperature, prompt, and sampling strategy, is **repeated five times**, and the average score is reported. The results for LLaMA3.1-8B-Instruct are presented in Table 3. The results in Table 3 reveal a consistent trend: At lower temperatures, top-H produces outputs with significantly higher creativity, originality, and coherence compared to min-$p$ and top-$p$ sampling methods in all three prompts. As $T$ increases, the top-$p$ sampling suffers a marked decline in coherence, often generating fragmented and incoherent text. This degradation stems from top-$p$'s lack of awareness of the model's confidence, as it truncates the distribution based purely on cumulative probability without accounting for distributional entropy.

In contrast, min-$p$ and top-H maintain stronger coherence at higher temperatures by adaptively limiting their sampling pools based on model confidence. Among the two, top-H consistently outperforms min-$p$ in both creativity and coherence. This is attributed to top-H's direct control over the entropy of the selected token set, allowing it to modulate randomness in alignment with the model's uncertainty. Additional LLM-as-a-Judge results supporting these conclusions are provided in Table 5 in the Appendix, which covers the evaluations on the Qwen2.5 and Phi-3-Mini models.

| Temperature | Prompt | Sampling | M1 | M2 | M3 | M4 | M5 | Average |
|---|---|---|---|---|---|---|---|---|
| 1.0 | Prompt 1 | Top-$p$ | 7.45 ±0.20 | 6.35 ±0.22 | **8.75** ±0.26 | 6.60 ±0.30 | 7.55 ±0.15 | 7.45 ±0.30 |
| | | Min-$p$ | 8.25 ±0.22 | 7.60 ±0.26 | 8.25 ±0.15 | 7.65 ±0.20 | 7.60 ±0.15 | 7.85 ±0.26 |
| | | Top-H | **8.80** ±0.15 | **8.65** ±0.20 | 8.40 ±0.22 | **8.05** ±0.15 | **8.55** ±0.22 | **8.45** ±0.26 |
| | Prompt 2 | Top-$p$ | 7.85 ±0.15 | 7.20 ±0.15 | **8.35** ±0.22 | 7.05 ±0.35 | 7.50 ±0.15 | 7.60 ±0.30 |
| | | Min-$p$ | 7.25 ±0.24 | 7.20 ±0.15 | 8.05 ±0.26 | 7.55 ±0.20 | 6.75 ±0.35 | 7.40 ±0.22 |
| | | Top-H | **8.10** ±0.2 | **8.10** ±0.3 | 8.25 ±0.15 | **7.90** ±0.15 | **8.35** ±0.26 | **8.25** ±0.30 |
| | Prompt 3 | Top-$p$ | 6.80 ±0.26 | 6.10 ±0.22 | **8.90** ±0.22 | 7.05 ±0.20 | 7.65 ±0.35 | 7.20 ±0.15 |
| | | Min-$p$ | 6.7 ±0.26 | 6.65 ±0.25 | 7.65 ±0.26 | 6.77 ±0.25 | 7.00 ±0.20 | 6.90 ±0.30 |
| | | Top-H | **8.15** ±0.26 | **8.05** ±0.26 | 8.00 ±0.15 | **8.30** ±0.31 | **8.25** ±0.20 | **8.05** ±0.30 |
| 1.5 | Prompt 1 | Top-$p$ | 7.45 ±0.15 | 7.10 ±0.26 | 8.20 ±0.22 | 7.55 ±0.30 | 7.35 ±0.22 | 7.55 ±0.31 |
| | | Min-$p$ | 7.95 ±0.22 | 7.55 ±0.35 | 8.25 ±0.20 | 7.55 ±0.26 | 7.60 ±0.22 | 7.80 ±0.26 |
| | | Top-H | **8.75** ±0.22 | **9.05** ±0.26 | **8.50** ±0.15 | **8.40** ±0.15 | **8.80** ±0.20 | **8.80** ±0.22 |
| | Prompt 2 | Top-$p$ | 7.80 ±0.30 | 7.75 ±0.22 | **8.65** ±0.35 | 7.00 ±0.22 | 7.65 ±0.26 | 7.75 ±0.22 |
| | | Min-$p$ | 7.30 ±0.20 | 7.10 ±0.31 | 7.87 ±0.30 | 6.80 ±0.26 | 6.75 ±0.26 | 7.10 ±0.15 |
| | | Top-H | **8.10** ±0.20 | **8.10** ±0.26 | 8.05 ±0.15 | **7.70** ±0.20 | **8.05** ±0.22 | **8.10** ±0.22 |
| | Prompt 3 | Top-$p$ | 7.40 ±0.20 | 6.85 ±0.26 | 7.70 ±0.15 | 7.20 ±0.22 | 8.35 ±0.31 | 7.45 ±0.30 |
| | | Min-$p$ | 6.35 ±0.20 | 6.05 ±0.22 | **7.85** ±0.22 | 6.55 ±0.15 | 7.20 ±0.26 | 6.80 ±0.26 |
| | | Top-H | **8.35** ±0.30 | **7.80** ±0.31 | 7.80 ±0.20 | **8.05** ±0.22 | **8.10** ±0.30 | **8.05** ±0.26 |
| 2.0 | Prompt 1 | Top-$p$ | 7.00 ±0.26 | 6.45 ±0.30 | 5.35 ±0.26 | 5.40 ±0.26 | 5.60 ±0.24 | 5.95 ±0.22 |
| | | Min-$p$ | 8.05 ±0.31 | 8.35 ±0.31 | 7.65 ±0.24 | 7.15 ±0.22 | 7.65 ±0.31 | 7.70 ±0.20 |
| | | Top-H | **8.80** ±0.22 | **9.05** ±0.24 | **8.75** ±0.26 | **8.70** ±0.15 | **8.80** ±0.22 | **8.85** ±0.20 |
| | Prompt 2 | Top-$p$ | 8.25 ±0.20 | 7.65 ±0.30 | 3.85 ±0.20 | 5.20 ±0.24 | 6.30 ±0.22 | 6.25 ±0.15 |
| | | Min-$p$ | 7.60 ±0.15 | 7.45 ±0.30 | 7.15 ±0.31 | 7.55 ±0.20 | 7.40 ±0.35 | 7.40 ±0.26 |
| | | Top-H | **8.85** ±0.20 | **8.35** ±0.31 | **8.60** ±0.26 | **8.60** ±0.30 | **8.70** ±0.22 | **8.60** ±0.22 |
| | Prompt 3 | Top-$p$ | 7.15 ±0.25 | **8.05** ±0.24 | 4.20 ±0.24 | 5.65 ±0.20 | **7.80** ±0.31 | 6.55 ±0.30 |
| | | Min-$p$ | 6.80 ±0.31 | 6.75 ±0.31 | 7.20 ±0.20 | 6.30 ±0.15 | 6.35 ±0.20 | 6.65 ±0.20 |
| | | Top-H | **8.0** ±0.31 | 7.1 ±0.22 | **9.0** ±0.15 | **8.05** ±0.20 | 7.05 ±0.20 | **7.65** ±0.24 |

Table 3: Evaluation metrics and the judge scores (on a scale of 1.0 to 10.0) for different temperatures, prompts, and sampling methods on **LLaMA3.1-8B-Instruct**. M1-M5 denote creativity, originality, narrative flow, imagery, and vitality, respectively.

In Appendix C, we present additional results and discussions on top-H, including **evaluations with a larger 70B model**, **human evaluation** of creativity and coherence across different sampling techniques, and **comparison of top-H to Mirostat method**.

### 6.1.4 Computational Overhead and Timing Comparisons

We compare per-token decode latency (ms/token) of top-H against top-$p$ and min-$p$ on three models: **LLaMA3.1-8B-Instruct**, **Phi-3-Mini-3.8B**, and **LLaMA3.3-70B-Instruct** in the Table 4. For each

configuration, we evaluate on 100 prompts from AlpacaEval, generating 128 tokens per prompt, and report the mean ms/token over prompts. Specifically, we observe a negligible overhead of **as low as 0.8%** compared to min-$p$ and top-$p$.

**Computational complexity.** Let $n$ denote the vocabulary size, and let $p_1 \geq p_2 \geq \cdots \geq p_n$ be the sorted probabilities. Sorting the logits dominates the computational cost for all cumulative decoding methods, requiring $O(n \log n)$ time. Subsequent operations such as partial selection or cumulative thresholding in top-$p$ and min-$p$ decoding only involve a single linear pass, adding $O(n)$ additional work but not changing the overall asymptotic complexity.

For top-H decoding, define the *partial entropy* $h_j = \sum_{i=1}^{j} p_i \log p_i$. According to the proof of Theorem A.3, the entropy of the distribution $q^j$ is given by

$$H(q^j) = \log \Gamma_j - \frac{h_j}{\Gamma_j},$$

where the cumulative mass satisfies $\Gamma_j = \Gamma_{j-1} + p_j$, and the partial entropy follows $h_j = h_{j-1} + p_j \log p_j$. These recurrences enable *incremental entropy accumulation* (Alg. 2), which updates $H(q^j)$ in $O(1)$ time per step, or $O(n)$ in total given sorted inputs. Therefore, the overall complexity of top-H decoding is also bounded by the sorting step, i.e., $O(n \log n)$. In practice, $\log p_j$ values are directly available from the model's log-probabilities.

---

**Algorithm 2** Incremental entropy accumulation

1: Initialize $\Gamma \leftarrow 0$, $h \leftarrow 0$, $H \leftarrow 0$
2: **for** each step $j$ **do**
3: $\quad \Gamma \leftarrow \Gamma + p_j$
4: $\quad h \leftarrow h + p_j \log p_j$
5: $\quad H \leftarrow \log(\Gamma) - \dfrac{h}{\Gamma}$
6: **end for**

---

| Temperature | LLaMA3.1-8B-Instruct | | | Phi-3-Mini | | | LLaMA3.3-70B-Instruct | | |
|---|---|---|---|---|---|---|---|---|---|
| | **Top-H** | **Min-$p$** | **Top-$p$** | **Top-H** | **Min-$p$** | **Top-$p$** | **Top-H** | **Min-$p$** | **Top-$p$** |
| 1.0 | 28.3951 | 27.3396 | 27.4275 | 24.3847 | 23.6499 | 23.7809 | 219.3837 | 219.1391 | 218.4900 |
| 2.0 | 28.4671 | 27.3840 | 27.4389 | 24.5929 | 23.9397 | 23.5844 | 219.3428 | 218.3609 | 217.7083 |

Table 4: Average runtime per token (ms/token) across sampling strategies and models.

## 6.2 Discussions and Ablations

**Sensitivity of the text to the temperature scaling.** In this section, we present a quantitative analysis of how the coherence of generated text varies with changes in sampling temperature. We conducted experiments using the Qwen2.5-3B and LLaMA3.1-8B-Instruct models on prompts from the Alpaca-Eval dataset. To operationalize coherence, we use the total log-probability (log-likelihood) of the generated sequence as a proxy: higher total log-probability suggests that the model is more confident in the output, which we interpret as a signal of greater coherence.

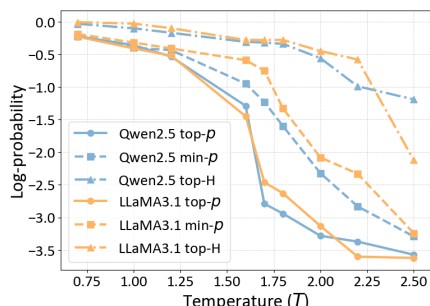

Figure 3: Effect of $T$ scaling on generation coherence in min-$p$, top-$p$ vs top-H.

Specifically, we compute the log-likelihood of each generated token during autoregressive generation, average these values across the entire sequence. This process is repeated in multiple temperature settings —0.7, 1.2, 1.6, 2.0, and 2.5—and for three different sampling strategies: top-$p$, min-$p$, and top-H. The result is portrayed in Fig. 3. As the temperature increases, the log-likelihood of the text generated under min-$p$ and top-$p$ sampling declines sharply. This suggests that the coherence of these methods is highly sensitive to temperature and that at higher temperatures, where increased creativity is encouraged, the generated

text tends to become less coherent. In contrast, top-H adjusts adaptively to the entropy of the distribution of the next token $H(p)$, effectively constraining randomness. As a result, it maintains more consistent and coherent output even in high-temperature settings.

**Impact of $\alpha$ parameter.** The only hyperparameter in top-H sampling is $\alpha$, which directly controls the maximum allowable entropy for the distribution $q$. As such, a careful tuning of $\alpha$ is essential. To determine an appropriate value, we randomly select 50 development samples from the Alpaca-Eval dataset and use LLaMA3.1–8B–Instruct to generate responses. We explore values of $\alpha$ in the range [0.1, 0.9], with increments of 0.05. For each candidate value, we run the model on the development set and evaluate the outputs using our LLM-as-a-judge prompt (the same as in Section 6.1.3) to assess both creativity and coherence. The optimal value of $\alpha$ is selected based on its ability to best balance these two objectives. The results of the creativity and coherence evaluation, averaged over 50 development samples, are presented in Figure 4. As $\alpha$ increases, the entropy threshold becomes more permissive, allowing greater randomness in token selection. Consequently, creativity tends to increase, while coherence tends to decline. The optimal value of $\alpha$ is the point at which these two metrics are best balanced. Based on the figure, we observe that $\alpha = 0.4$ produces the highest average in the creativity and coherence scores, indicating it as the most suitable choice. Additional quantitative results are provided in Appendix C.5.

**Empirical optimality of the top-H decoding strategy.** We now empirically evaluate the competitiveness of the greedy algorithm of the top-H relative to the optimal solution of the ECMM problem, found by exhaustive search. We randomly sample 20 prompts from the Alpaca-Eval dataset and generate responses using the top-H method. At each generation step, the candidate set of tokens is restricted to the top-15 tokens of the probability distribution predicted by the model. To obtain the optimal solution, we exhaustively enumerate all possible $2^{15}$ subsets of the feasible token set and identify the subset $S^*$ that maximizes the objective $\Gamma_{S^*}$, subject to the entropy constraint $H(q) \leq 0.4\,H(p)$, with $q$ denoting the distribution over selected subset.

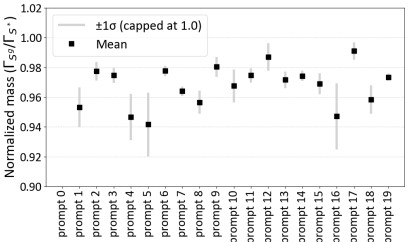

Figure 4: Effect of the parameter $\alpha$ on creativity and coherence.

For comparison, we also compute the greedy solution $S^g$ using Algorithm 1. At each generation step, we calculate the ratio $\Gamma_{S^g}/\Gamma_{S^*}$, and report the mean and variance of this ratio (across different generation steps) for 20 different evaluation prompts, as visualized in Figure 5. As shown in the figure, the mean of the ratio remains consistently close to 1.0 across randomly sampled instances from the dataset, with only minor variance. Although deriving a formal approximation guarantee is beyond the scope of this work, our empirical results indicate that the solution obtained by top-H for the ECMM problem closely approximates the optimal solution in practice.

Figure 5: Empirical evaluation of top-H performance relative to the optimal solution of the ECMM problem.

While the primary goal of this work is to empirically demonstrate the efficacy of top-H decoding in addressing the ECMM, we defer a detailed theoretical analysis of the associated error bounds to future research. Nevertheless, in Appendix A.4, we provide a preliminary worst-case error bound for the greedy top-H solution under a specific assumption about the next-token probability distribution.

# 7 Conclusions

This paper addresses the challenge of balancing creativity and coherence in LLMs, particularly under high-temperature settings, where coherence often deteriorates. We introduce the entropy-constrained mass maximization (ECMM) problem, which formalizes the objective of balancing creativity and coherence by imposing an entropy constraint on the distribution of tokens in the sampling set. After proving the NP-hardness of ECMM, we propose top-H, a computationally efficient greedy algorithm that effectively approximates the solution of ECMM problem. Extensive empirical evaluation across various tasks demonstrates that top-H consistently outperforms established sampling strategies such as top-$p$ and min-$p$, achieving up to **25.6%** higher accuracy. These results establish top-H as a new state-of-the-art method for creative writing in LLMs.

## Acknowledgments

This work was partially supported by a grant from the Directorate for Computer and Information Science and Engineering (CISE) of the National Science Foundation.

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

# A Proofs of Theorems

Unless otherwise specified, all logarithms are natural log.

## A.1 Proof of Theorem 1

*Proof of Theorem 1.* The probability of the selected tokens needs to be divided by their sum, to make sure that the sum of the distribution q is 1. Given:

$$\sum_i p_i = 1 \quad \Rightarrow \quad \sum_i p_i \mathbb{1}_{\{v_i \notin S\}} = 1 - \Gamma_S \tag{6}$$

According to 4:

$$M = \begin{cases} \frac{p_i + \frac{p_i}{\Gamma_S}}{2} & i \le m \\[2mm] \frac{p_i}{2} & i > m \end{cases}$$

$$D_{KL}(p||M) = \sum_{i=1}^{m} p_i \log\left(\frac{p_i}{\frac{1}{2}\left(p_i + \frac{p_i}{\Gamma_S}\right)}\right) + \sum_{i=m+1}^{n} p_i \log\left(\frac{p_i}{\frac{1}{2}p_i}\right) = \sum_{i=1}^{m} p_i \log\left(\frac{2\Gamma_S}{1+\Gamma_S}\right) + \sum_{i=m+1}^{n} p_i \log(2)$$

$$= \log\left(\frac{2\Gamma_S}{1+\Gamma_S}\right) \sum_{i=1}^{m} p_i + \log(2) \sum_{i=m+1}^{n} p_i$$

Using (3) and (6):

$$= \Gamma_S \log\left(\frac{2\Gamma_S}{1+\Gamma_S}\right) + \log(2)(1 - \Gamma_S) = \Gamma_S\left[\log\left(\frac{\Gamma_S}{1+\Gamma_S}\right) + \log(2)\right] + \log(2)(1 - \Gamma_S)$$

$$= \Gamma_S \log\left(\frac{\Gamma_S}{1+\Gamma_S}\right) + \log(2)\Gamma_S + \log(2) - \log(2)\Gamma_S = \Gamma_S \log\left(\frac{\Gamma_S}{1+\Gamma_S}\right) + \log(2) \tag{7}$$

$$D_{KL}(q||M) = \sum_{i=1}^{m} \frac{p_i}{\Gamma_S} \log\left(\frac{\frac{p_i}{\Gamma_S}}{\frac{1}{2}\left(p_i + \frac{p_i}{\Gamma_S}\right)}\right) = \frac{1}{\Gamma_S} \sum_{i=1}^{m} p_i \log\left(\frac{2}{1+\Gamma_S}\right) = \frac{1}{\Gamma_S} \log\left(\frac{2}{1+\Gamma_S}\right) \sum_{i=1}^{m} p_i$$

Using (3):

$$= \log\left(\frac{2}{1+\Gamma_S}\right) = \log(2) - \log(1 + \Gamma_S) \tag{8}$$

Rewriting Jensen-Shannon Divergence using (7) and (8):

$$\mathrm{JSD}(p||q) = \frac{1}{2}\left(\log(2) + \Gamma_S \log\left(\frac{\Gamma_S}{1+\Gamma_S}\right)\right) + \frac{1}{2}\left(\log(2) - \log(1 + \Gamma_S)\right)$$

$$= \frac{1}{2}\left(2\log(2) + \Gamma_S \log(\Gamma_S) - \Gamma_S \log(1 + \Gamma_S) - \log(1 + \Gamma_S)\right) = \boxed{\log(2) + \frac{1}{2}\left(\Gamma_S \log(\Gamma_S) - (1 + \Gamma_S)\log(1 + \Gamma_S)\right)}$$

Therefore, JSD is only dependent on $\Gamma_S$.
Now we want to show that the distance between the distributions is decreasing with respect to $\Gamma_S$:

$$\frac{d}{d\Gamma_S}\mathrm{JSD}(p||q) = \frac{1}{2}\left(\log(\Gamma_S) + 1 - \log(1 + \Gamma_S) - 1\right)$$

$$= \frac{1}{2}\left(\log(\Gamma_S) - \log(1 + \Gamma_S)\right) \quad \rightarrow \quad \text{Always negative.}$$

Therefore, $\mathrm{JSD}(p||q)$ is decreasing with respect to $\Gamma_S$ and to minimize $\mathrm{JSD}(p||q)$, one needs to maximize $\Gamma_S$. $\qquad\square$

## A.2 NP-Hardness of Entropy-Constrained Mass Maximization

All logarithms are natural ($\ln$). Arithmetic is performed on a *unitcost RAM with binary encodings*; an integer $x \geq 1$ occupies $\lfloor \log_2 x \rfloor + 1$ bits.

### A.2.1 Problem definition

For a probability vector $\mathbf{p} = (p_1, \ldots, p_n)$ with $\sum_i p_i = 1$, define

$$H(\mathbf{p}) := -\sum_{i=1}^{n} p_i \ln p_i.$$

The fixed–budget maximization problem ECMM is

$$\max_{S \subseteq [n]} \Gamma_S := \sum_{i \in S} p_i \quad \text{s.t.} \quad H(S) \leq 0.4 \cdot H(\mathbf{p}), \tag{ECMM}$$

where $H(S)$ denotes the entropy of the renormalized vector $(p_i)_{i \in S}$.

The corresponding decision version, ECME, is:

> *Input:* A probability vector $\mathbf{p} = (p_1, \ldots, p_n)$, a mass target $\beta = \frac{2}{3}$, and the fixed budget $\alpha = 0.4 \cdot H(\mathbf{p})$.
> *Question:* Does there exist $S \subseteq [n]$ such that
> $$\sum_{i \in S} p_i = \beta \quad \text{and} \quad H(S) \leq \alpha?$$

We show that the decision variant is NP–complete; the optimization variant is NP–hard.

### A.2.2 Source problem: Cardinality–Constrained Subset Sum

**Definition 1** (CCSS). *Given positive integers $w_1, \ldots, w_m$, a target $\tau$, and an integer $K$ with $3 \leq K \leq m$, decide whether some subset of exactly $K$ weights sums to $\tau$.*

CCSS is NP–complete: reduce from the classic SUBSET–SUM; see, e.g., Papadimitriou (Papadimitriou, 1994, Exercise 8.14).

Our reduction from CCSS to (ECMM) needs the following *narrow–range* condition.

**Assumption 1** (Narrow range).

$$\frac{\tau}{K+1} < w_i < \frac{\tau}{K-1} \quad (1 \leq i \leq m).$$

The next lemma shows that we may enforce Assumption 1 by a *polynomial–time padding* step.

**Lemma 1** (Padding to narrow range). *There is a polynomial–time transformation that maps an arbitrary CCSS instance $(w_1, \ldots, w_m; \tau; K)$ to an equivalent instance $(w'_1, \ldots, w'_m; \tau'; K)$ that satisfies Assumption 1. The new weights and target have binary lengths polynomial in the original instance size.*

*Proof.* Set
$$M := (K+1)\,\tau, \qquad w'_i := w_i + M, \qquad \tau' := \tau + K\,M.$$
Because the same constant $M$ is added to every weight, a subset of exactly $K$ items sums to $\tau$ iff it sums to $\tau'$:
$$\sum_{i \in S} w_i = \tau \iff \sum_{i \in S} w'_i = \tau'.$$

**Lower bound.** Each new weight satisfies $w'_i > M$. Moreover
$$\frac{\tau'}{K+1} = \frac{\tau + K(K+1)\tau}{K+1} = KT + \frac{\tau}{K+1} < K\tau + \tau = M,$$
so $w'_i > \tau'/(K+1)$.

**Upper bound.** We have $w_i' \leq w_{\max} + M \leq \tau + (K+1)\tau = (K+2)\tau$. On the other hand,

$$\frac{\tau'}{K-1} = \frac{(K(K+1)+1)\tau}{K-1} = \frac{K^2+K+1}{K-1}\tau.$$

Since $(K+2)(K-1) = K^2+K-2 < K^2+K+1$ for every $K \geq 3$, it follows that $w_i' < \tau'/(K-1)$. Therefore Assumption 1 holds for the padded instance.

**Encoding size.** The multiplier $M = (K+1)\tau$ increases the bit–length of the largest weight by at most $\log_2(K+1)$ bits, and the same holds for $\tau'$. Hence the transformation is polynomial in the input length. $\qquad\square$

Henceforth we assume without loss of generality that every CCSS instance meets Assumption 1; if it does not, we first apply the padding from Lemma 1.

We also stipulate $K \geq 20$ (duplicate the instance as below if necessary).

**Scaling step (making $K \geq 20$ while keeping the narrow range).** If the given instance has $K < 20$, put

$$d := \lceil 20/K \rceil,$$

duplicate *every* weight $d$ times and set

$$K_1 := dK, \qquad \tau_1 := d\tau.$$

Then

$$\exists S \subseteq [m] : |S| = K, \sum_{i \in S} w_i = \tau \iff \exists S \subseteq [m] : |S| = K_1, \sum_{i \in S} w_i = \tau_1,$$

so feasibility is preserved and $K_1 \geq 20$.

The plain duplication, however, may violate Assumption 1 because the new lower bound $\tau_1/(K_1+1)$ can exceed some of the duplicated weights. To restore the assumption we now *re-apply* the padding of Lemma 1 to the instance $(w_1, \ldots, w_m; \tau_1; K_1)$. This yields an equivalent instance

$$(w_1', \ldots, w_m'; \tau'; K_1)$$

that *satisfies* Assumption 1 and still has $K_1 \geq 20$.

All numbers grow by at most $\lceil \log_2 d \rceil + O(\log K)$ bits, so the whole transformation remains polynomial-time.

Henceforth we may—and do—assume that $K \geq 20$ and that the narrow-range condition holds.

### A.2.3  Reduction to ECME

Let $(w_1, \ldots, w_m; \tau; K)$ be a CCSS instance that already satisfies Assumption 1. Define

$$\gamma_K := \frac{1}{16K^2}, \quad \theta_K < \frac{1}{2K^2}, \quad \delta_K := \frac{5\theta_K}{2\ln K}, \quad \varepsilon_K := \frac{0.0384 + \gamma_K}{\ln K},$$

$$\lambda(K) := \left\lceil \frac{0.7333 - \varepsilon_K + \delta_K}{0.133} \right\rceil, \quad B := \lceil K^{\lambda(K)} \rceil,$$

$$w_b := \frac{\tau}{2B}, \quad \tau_b := Bw_b = \tfrac{1}{2}\tau, \quad W := \tau + \tau_b = \tfrac{3}{2}\tau.$$

Set

$$p_i := \frac{w_i}{W} \quad (1 \leq i \leq m), \quad p_b := \frac{w_b}{W} = \frac{1}{3B}, \quad \beta := \frac{\tau}{W} = \tfrac{2}{3}.$$

The resulting ECME instance contains $n = m + B$ items. The numbers above are representable with $O(\log K)$ bits, hence the reduction runs in polynomial time.

### A.2.4 Entropy budget window

**Lemma 2** (Budget window). *For the constructed instance,*

$$\ln K - \gamma_K < 0.4\, H(\mathbf{p}) < \ln(K+1).$$

*Proof.* Split $H(\mathbf{p}) = H_{\mathrm{h}} + H_{\mathrm{b}}$, where

$$H_{\mathrm{h}} = \tfrac{2}{3}\bigl(H(\mathbf{q}) + \ln\tfrac{2}{3}\bigr), \quad q_i := \frac{w_i}{\tau},$$

$$H_{\mathrm{b}} = \tfrac{1}{3}\bigl(\ln 3 + \lambda(K)\ln K\bigr).$$

By Assumption 1 and a second–order Taylor bound, $H(\mathbf{q}) = \ln K - \theta_K$ with $0 < \theta_K < \frac{1}{2K^2}$. Substitution gives

$$0.4\, H(\mathbf{p}) = (1 - \varepsilon_K)\ln K + 0.0384 - 0.2667\,\theta_K + 0.133\,\delta_K \ln K = \ln K - \gamma_K + 0.0666\,\theta_K,$$

and since $0 < 0.0666\,\theta_K < \ln(1 + 1/K)$ for $K \geq 20$, the window follows. $\square$

### A.2.5 Structural lemmas

**Lemma 3** (Booster blow-up). *Let $S$ be any subset with $\Gamma_S = \beta$. If $S$ contains at least one booster item, then $H(S) > 0.4\, H(\mathbf{p})$.*

*Proof.* Write $S = H \cup B$ where $H$ (resp. $B$) is the set of heavy (resp. booster) indices selected. Let $b := |B| \geq 1$ and $L := |H|$.

**Step 1 – how many boosters are needed.** Each booster weighs $w_b = \tau/(2B)$, whereas every heavy weight is at least $w_{\min} = \tau/(K+1)$ by Assumption 1. Total weight has to be exactly $\tau$, so each heavy item that is *removed* must be replaced by at least

$$\frac{w_{\min}}{w_b} = \frac{\tau/(K+1)}{\tau/(2B)} = \frac{2B}{K+1} > \frac{2B}{K}$$

boosters. Therefore $L \leq K - 1$ and

$$b \;\geq\; \frac{2B}{K}, \qquad \delta := \frac{b}{2B} \;\geq\; \frac{1}{K}. \tag{9}$$

(The quantity $\delta$ equals the total probability mass of the boosters after renormalisation because each has probability $w_b/\tau = 1/(2B)$.)

**Step 2 – a lower bound on the entropy of $S$.** The booster probabilities are all $1/(2B)$, so their contribution is $\delta \ln(2B)$. For the heavy part we use the crude bound $\ln(K - 1) \leq \ln K - 1/K$ together with the fact that the $L$ heavy probabilities add up to $1 - \delta$:

$$H(S) \;\geq\; (1 - \delta)\ln(K - 1) + \delta \ln(2B) \;\geq\; \ln K - \frac{1}{K} + \delta\bigl((\lambda(K) - 1)\ln K + \ln 2\bigr),$$

because $\ln(2B) = \ln 2 + \lambda(K)\ln K$ by definition of $B$. With (9) this gives

$$H(S) \;\geq\; \ln K \;+\; \frac{(\lambda(K) - 1)\ln K + \ln 2 - 1}{K}.$$

Since $\lambda(K) \geq 2$ for every $K \geq 20$, the numerator is positive and we conclude

$$H(S) > \ln K. \tag{10}$$

**Step 3 – compare with the budget.** Lemma 2 states $0.4\, H(\mathbf{p}) < \ln K$. Combining this with (10) proves $H(S) > 0.4\, H(\mathbf{p})$, as required. $\square$

**Lemma 4** (Cardinality lock). *If $S$ contains no boosters and $\Gamma_S = \beta = 2/3$, then $|S| = K$ and $\sum_{i \in S} w_i = \tau$.*

*Proof.* Since $S$ has no boosters, its total weight is

$$\sum_{i \in S} w_i = W \, \Gamma_S = \tau.$$

Let $|S| = L$. By the narrow–range assumption,

$$L \cdot \frac{\tau}{K+1} < \sum_{i \in S} w_i = \tau < L \cdot \frac{\tau}{K-1}.$$

Dividing through by $\tau$ gives

$$\frac{L}{K+1} < 1 < \frac{L}{K-1},$$

which simplifies to $K - 1 < L < K + 1$. Since $L$ is an integer, $L = K$. Having $|S| = K$ and total weight $\tau$ establishes the claim. $\qquad\square$

**Lemma 5** (Entropy gap for a $K$–heavy subset). *Every $K$–element subset $S$ of heavy items summing to $\tau$ satisfies*

$$H(S) \leq \ln K - \gamma_K.$$

*Proof.* After selecting $K$ heavy items summing to $\tau$, their renormalised probabilities are $r_i = w_i / \tau$. By the narrow–range assumption,

$$\frac{1}{K+1} < r_i < \frac{1}{K-1},$$

so we may write $r_i = \frac{1}{K} + x_i$ with $|x_i| < \frac{1}{K(K-1)}$ and $\sum_i x_i = 0$.

Then

$$H(S) = -\sum_{i=1}^{K} r_i \ln r_i = -\sum_{i=1}^{K} \left(\tfrac{1}{K} + x_i\right) \ln\left(\tfrac{1}{K} + x_i\right) = \ln K - \sum_{i=1}^{K} \left(\tfrac{1}{K} + x_i\right) \ln\left(1 + K x_i\right).$$

Using the Taylor approximation $\ln(1 + u) \geq u - \frac{u^2}{2}$ for $|u| < 1$,

$$\left(\tfrac{1}{K} + x_i\right) \ln(1 + K x_i) \geq \left(\tfrac{1}{K} + x_i\right)\left(K x_i - \tfrac{(K x_i)^2}{2}\right) = x_i + \tfrac{K x_i^2}{2} - K^2 x_i^3 / 2.$$

Summing over $i$ and using $\sum_i x_i = 0$ and $|x_i| < 1/(K(K-1))$ gives

$$\sum_{i=1}^{K} \left(\tfrac{1}{K} + x_i\right) \ln(1 + K x_i) \geq \frac{K}{2} \sum_{i=1}^{K} x_i^2 - \frac{K^2}{2} \sum_{i=1}^{K} |x_i|^3 > \frac{1}{2(K-1)^2} - \frac{1}{2(K-1)^3} = \frac{K-2}{2(K-1)^3}.$$

For $K \geq 20$, one checks

$$\frac{K-2}{2(K-1)^3} > \frac{1}{16K^2} = \gamma_K.$$

Hence

$$H(S) = \ln K - \sum_i \left(\tfrac{1}{K} + x_i\right) \ln(1 + K x_i) \leq \ln K - \gamma_K,$$

as claimed. $\qquad\square$

### A.2.6 Equivalence theorem

**Theorem 4.** *The constructed* ECME *instance admits a subset of mass $\beta$ iff the original* CCSS *instance is a YES instance.*

*Proof.* ($\Rightarrow$) Any feasible $S$ must exclude boosters by Lemma 3, so Lemma 4 gives a $K$–subset summing to $\tau$.
($\Leftarrow$) Conversely, let $S$ be any $K$–element subset summing to $\tau$. By Lemma 5, $H(S) \leq \ln K - \gamma_K$, and by Lemma 2, $\ln K - \gamma_K < 0.4\, H(\mathbf{p})$. Hence $H(S) < 0.4\, H(\mathbf{p})$ and $\Gamma_S = \beta$, so $S$ is feasible. $\qquad\square$

### A.2.7   Complexity consequence

**Theorem 5.** *The decision variant* ECME *is* **NP–complete**, *and the corresponding optimization problem* ECMM *is* **NP–hard**.

*Proof.* **Membership in NP** Given a candidate subset $S$, we can verify both constraints in polynomial time. For the mass we simply add the $p_i$'s. For the entropy, we approximate each ln to $O(\log K)$ bits; the required precision is well below the separating gap $\gamma_K - 0.0666\,\theta_K = \Theta(K^{-2})$, so rounding cannot flip the inequality. Classical results on transcendental evaluation on a unit–cost RAM (Brent & Zimmermann, 2010) show that such an approximation takes $\tilde{O}((\log K)^2)$ time—polynomial in the input size. Hence the verifier runs in polynomial time.

**NP–hardness of the decision problem.** Apply the polynomial–time padding from Lemma 1 and then the reduction of Section A.2.3. By Theorem 4, the resulting instance is a *YES* instance of ECME iff the original CCSS instance is a *YES* instance. Therefore the decision problem is NP–hard.

**Optimization hardness.** Assume, for contradiction, that we had a polynomial–time algorithm that returns

$$\max_{S\subseteq[n]} \Gamma_S \quad \text{s.t.} \quad H(S) \le 0.4 \cdot H(\mathbf{p}).$$

On the same input we could decide the ECME instance by a single comparison of that maximum with the fixed target value $\beta = \frac{2}{3}$. This would solve an NP–complete problem in polynomial time, contradicting P $\neq$ NP. Hence the optimization version is **NP–hard**. $\square$

### A.3   Proof of Early Termination

*Proof of Theorem 3.* Assume that the distribution of the selected tokens after j steps is $q^j$, therefore:

$$H(q^{j-1}) = -\sum_{i=1}^{j-1} \frac{p_i}{\Gamma_{j-1}} \log\left(\frac{p_i}{\Gamma_{j-1}}\right) = -\frac{1}{\Gamma_{j-1}}\sum_{i=1}^{j-1} p_i \log p_i + \frac{\log(\Gamma_{j-1})}{\Gamma_{j-1}}\sum_{i=1}^{j-1} p_i, \quad \text{where } \Gamma_{j-1} = \sum_{i=1}^{j-1} p_i$$

Since $\sum_{i=1}^{j-1} p_i = \Gamma_{j-1}$:

$$H(q^{j-1}) = \log(\Gamma_{j-1}) - \frac{1}{\Gamma_{j-1}}\sum_{i=1}^{j-1} p_i \log p_i$$

$$\therefore -\sum_{i=1}^{j-1} p_i \log p_i = \Gamma_{j-1}(H(q^{j-1}) - \log(\Gamma_{j-1})) \tag{11}$$

Now, calculating $H(q^j)$:

$$H(q^j) = -\sum_{i=1}^{j} \frac{p_i}{\Gamma_{j-1}+p_j} \log\left(\frac{p_i}{\Gamma_{j-1}+p_j}\right) = -\sum_{i=1}^{j} \frac{p_i}{\Gamma_{j-1}+p_j}\log p_i + \sum_{i=1}^{j}\frac{p_i}{\Gamma_{j-1}+p_j}\log(\Gamma_{j-1}+p_j)$$

$$= -\frac{1}{\Gamma_{j-1}+p_j}\sum_{i=1}^{j} p_i \log p_i + \frac{\log(\Gamma_{j-1}+p_j)}{\Gamma_{j-1}+p_j}\sum_{i=1}^{j} p_i$$

Since $\sum_{i=1}^{j} p_i = \sum_{i=1}^{j-1} p_i + p_j = \Gamma_{j-1}+p_j$:

$$H(q^j) = \log(\Gamma_{j-1}+p_j) - \frac{1}{\Gamma_{j-1}+p_j}\sum_{i=1}^{j} p_i \log p_i = \log(\Gamma_{j-1}+p_j) + \frac{1}{\Gamma_{j-1}+p_j}\left(-\sum_{i=1}^{j-1} p_i \log p_i - p_j \log p_j\right)$$

Using (11):

$$= \log(\Gamma_{j-1} + p_j) + \frac{1}{\Gamma_{j-1} + p_j} \left( \Gamma_{j-1} H(q^{j-1}) - \Gamma_{j-1} \log(\Gamma_{j-1}) - p_j \log p_j \right)$$

$$\therefore \Delta H = H(q^j) - H(q^{j-1})$$

$$= \log(\Gamma_{j-1} + p_j) + \frac{1}{\Gamma_{j-1} + p_j} \left( \Gamma_{j-1} H(q^{j-1}) - \Gamma_{j-1} \log(\Gamma_{j-1}) - p_j \log p_j \right) - H(q^{j-1})$$

$$= \log(\Gamma_{j-1} + p_j) + \frac{1}{\Gamma_{j-1} + p_j} \left( \Gamma_{j-1} H(q^{j-1}) - (\Gamma_{j-1} + p_j) H(q^{j-1}) - \Gamma_{j-1} \log(\Gamma_{j-1}) - p_j \log(p_j) \right)$$

$$= \log(\Gamma_{j-1} + p_j) - \frac{1}{\Gamma_{j-1} + p_j} \left( p_j H(q^{j-1}) + \Gamma_{j-1} \log(\Gamma_{j-1}) + p_j \log(p_j) \right)$$

The probabilities are sorted in descending order, therefore:

$$\forall i \leq j : p_j \leq p_i \Rightarrow \sum_{i=1}^{j-1} p_j \leq \sum_{i=1}^{j-1} p_i \Rightarrow (j-1) p_j \leq \Gamma_{j-1} \Rightarrow j - 1 \leq \frac{\Gamma_{j-1}}{p_j}$$

$$\therefore \quad H(q^{j-1}) \leq \log(j-1) \leq \log\left( \frac{\Gamma_{j-1}}{p_j} \right)$$

$$\therefore \quad \Delta H \geq \log(\Gamma_{j-1} + p_j) - \frac{p_j \log\left( \frac{\Gamma_{j-1}}{p_j} \right)}{\Gamma_{j-1} + p_j} - \frac{\Gamma_{j-1} \log(\Gamma_{j-1}) + p_j \log(p_j)}{\Gamma_{j-1} + p_j}$$

$$= \log(\Gamma_{j-1} + p_j) - \frac{1}{\Gamma_{j-1} + p_j} \left[ p_j \log(\Gamma_{j-1}) - p_j \log(p_j) + \Gamma_{j-1} \log(\Gamma_{j-1}) + p_j \log(p_j) \right]$$

$$= \log(\Gamma_{j-1} + p_j) - \frac{1}{\Gamma_{j-1} + p_j} \left( p_j \log(\Gamma_{j-1}) + \Gamma_{j-1} \log(\Gamma_{j-1}) \right)$$

$$= \log(\Gamma_{j-1} + p_j) - \log(\Gamma_{j-1}) = \log\left( \frac{\Gamma_{j-1} + p_j}{\Gamma_{j-1}} \right) = \log\left( 1 + \frac{p_j}{\Gamma_{j-1}} \right)$$

$$\therefore \quad \Delta H \geq \log\left( 1 + \frac{p_j}{\Gamma_{j-1}} \right) > 0 \Rightarrow \Delta H > 0$$

$$\square$$

## A.4   Formal approximation bound for the top-H

**Zipf model.**

Fix a vocabulary of size $n$ and exponent $s > 1$. We assume the sorted next-token probabilities obey the classical Zipf / regularly varying law:

$$p_i := \frac{i^{-s}}{H_{n,s}}, \quad H_{n,s} = \sum_{j=1}^{n} j^{-s}. \tag{12}$$

Empirically, language-model logits are well approximated by $s \in [1.05, 1.20]$ (Gerlach & Altmann, 2013); hence the assumption captures current practice.

The Zipf assumption matters, because ECMM is NP-hard (Theorem 2), exact polynomial-time solutions are unlikely. In the absence of structural assumptions, a constant-factor approximation guarantee for ECMM is highly unlikely unless $P = NP$, as our NP-hardness proof relies on a gap-preserving reduction from Cardinality-Constrained Subset Sum. This structure is known to be fundamentally difficult to approximate, a standard result in computational complexity theory (Papadimitriou, 1994). However, LLM logits consistently follow heavy-tailed (Zipf / regularly-varying) laws, so analysing this regime yields practically relevant bounds.

**Notation.** We write $M(k) = \sum_{i \leq k} p_i$ for the prefix mass, $T(k) = 1 - M(k)$ for the tail mass, and $H(k)$ for the entropy of the normalised prefix distribution $q_i^{(k)} = p_i / M(k)$.

## Preliminaries

**Lemma 6** (Monotonicity of prefix entropy). *For $1 \leq k < n$ one has $H(k) < H(k+1)$. Moreover, for any subset $S$ of size $k$, $H(q_S) \geq H(k)$.*

*Proof.* The first claim is classical: adding the $(k+1)$-st symbol strictly increases entropy because $p_{k+1} < p_k$ and entropy is Schur-concave. For the second claim we note that $(p_1, \ldots, p_k)$ majorises any other $k$-subset of the sorted vector; Schur-concavity again yields the desired inequality. $\square$

**Lemma 7** (Tail mass bound). *For $k < n$,*

$$\frac{(k+1)^{1-s} - n^{1-s}}{(s-1)H_{n,s}} \leq T_n(k) \leq \frac{k^{1-s}}{(s-1)H_{n,s}}$$

*When $n \gg k$, the numerator difference is $o(k^{1-s})$, so the upper bound is asymptotically tight.*

*Proof.* Apply upper and lower Riemann sums to $\int_k^n x^{-s}\,dx$.

$\square$

**Lemma 8** (Prefix entropy asymptotics). *There exist constants $c_s, C_s > 0$ such that*

$$c_s + \frac{s}{s-1}\log\left(\frac{k}{2}\right) \leq H(k) \leq C_s + \frac{s}{s-1}\log k \quad \text{for all } k \geq 2.$$

*Proof.* Combine Lemma 7 with integral bounds for $\sum i^{-s} \log i$. $\square$

## Depth of the greedy prefix

Let $k_g := \max\{k : H(k) \leq \alpha H(n)\}$. Lemma 6 implies $k_g$ is well defined.

**Lemma 9** (Growth rate of $k_g$). *There exist constants $a_s, b_s > 0$ such that*
$$a_s n^\alpha \leq k_g \leq b_s n^\alpha.$$

*Proof.* Insert Lemma 8 into the defining inequality and solve for $k_g$. $\square$

## Mass captured by top-H

Write $\Gamma_g = M(k_g)$. This is the mass captured by the greedy solution. By construction, this solution is valid as it satisfies the entropy constraint $H(k_g) \leq \alpha H(n)$.

## Tight upper bound for ECMM

Let $S^\star$ be any subset satisfying the entropy constraint, and set $\Gamma^\star = \sum_{i \in S^\star} p_i$. We want to bound the maximum possible value of $\Gamma^\star$.

The greedy algorithm selects the prefix $[k_g]$ and captures a mass of $\Gamma_g = M(k_g) = 1 - T(k_g)$. The total mass of all tokens not in the greedy solution is, by definition, the tail mass $T(k_g)$.

Any other valid solution $S^\star$ can, at best, capture the mass of the greedy solution *plus* some portion of the remaining tail mass. The absolute maximum mass any solution can capture is 1 (the entire vocabulary). Therefore, the maximum possible improvement any optimal solution $\Gamma^\star$ can have over the greedy solution $\Gamma_g$ is bounded by the tail mass that the greedy algorithm left behind:

$$\Gamma^\star - \Gamma_g \leq 1 - \Gamma_g = 1 - M(k_g) = T(k_g). \tag{13}$$

This gives us a direct upper bound on the additive gap between the optimal and greedy solutions.

**Theorem 6** (Distribution-dependent additive guarantee). *Under equation 12, for every $n \geq 4$,*

$$\Gamma^\star - \Gamma_g \leq T_n(k_g) \leq \frac{k_g^{s-1}}{(s-1)H_{n,s}} = \mathcal{O}(n^{-\alpha(s-1)})$$

*Proof.* From 13, we have the additive gap $\Gamma^\star - \Gamma_g \leq T(k_g) = T_n(k_g)$. Lemma 7 bounds $T_n(k_g)$ by $\frac{k_g^{1-s}}{(s-1)H_{n,s}}$. Finally, lemma 9 gives $k_g = \Theta(n^\alpha)$, so the gap decays as $\mathcal{O}(n^{-\alpha(s-1)})$. $\square$

**Discussion on the effectiveness of greedy**

We understand that the approach of dropping high-probability tokens could in principle beat the greedy prefix by allowing more low-mass tokens from the tail. However, this problem (ECMM) is NP hard, and we approximate the solution via a practically feasible greedy approach. Notably, we empirically demonstrate the effectiveness of this greedy based top-H approach to be superior to the existing SoTA. Additionally, under the constrained distribution of the Zipfian regime, the greedy prefix is constructed to generally maximize mass under minimal entropy growth. Because entropy increases monotonically with each added token, and the most probable tokens usually contribute less to entropy per unit mass, the greedy approach reaches the entropy threshold more efficiently than any alternative.

# B   LLM-as-a-Judge Evaluation Prompts

Following (Nguyen et al., 2024), we adopt the following judge evaluation prompt and three open-ended prompts designed to elicit creative responses and facilitate creativity–coherence trade-off analysis:

---

**Judge Evaluation Prompt**

You are an expert judge evaluating AI-generated creative writing. I am testing the diversity and coherent writing capabilities of three different models. I will paste three different responses that were generated here. Rate responses based on the following metrics:
**1. Diversity:** Novelty and uniqueness of ideas   **2. Originality:** Innovative approach to the prompt   **3. Narrative Flow:** Coherence of the text   **4. Emotional Impact:** Ability to evoke feelings   **5. Imagery:** Vividness of descriptions.
Rate each metric from 1 to 10. Also, suggest the overall winner: the response that best maintains high coherence while demonstrating high diversity.

---

**Prompt 1**

Write a story about an alien civilization's first contact with Earth from their perspective.

---

**Prompt 2**

Write a story about a world where time suddenly starts moving backwards.

---

**Prompt 3**

Write a story about a mysterious door that appears in an unexpected place.

---

# C   More Results

In this section we provide more experimental evaluations and analysis on top-H sampling.

### C.1   LLM-as-a-Judge for creativity and coherence evaluation

Table 5 presents creativity and coherence evaluations under the LLM-as-a-Judge setup for the Qwen2.5–3B and Phi-3-Mini–4k–Instruct models. The observed trends are consistent with those of LLaMA3.1–8B–Instruct: as the decoding temperature increases, coherence degrades notably for both min-$p$ and top-$p$ sampling methods. In contrast, top-H effectively maintains coherence while producing more creative outputs.

### C.2   Validation with large model

We replicated the MT-Bench (Table 6) and GPQA (Table 7) experiments using **LLaMA3.3-70B-Instruct** with the setup of Section 6.1 in the paper. Specifically, top-H outperforms min-$p$ by up to 6.0% and 6.47% on MT-Bench and GPQA, respectively.

| LLM | Temperature | Prompt | Sampling | M1 | M2 | M3 | M4 | M5 |
|---|---|---|---|---|---|---|---|---|
| | | Prompt 1 | Top-$p$ | 7.4 | 7.4 | **8.0** | 8.0 | 7.2 |
| | | | Min-$p$ | 6.6 | 6.5 | 7.8 | 6.7 | 7.2 |
| | | | Top-H | **8.4** | **8.4** | 8.0 | **8.1** | **8.6** |
| | 1.0 | Prompt 2 | Top-$p$ | 7.2 | 7.0 | **8.4** | 6.6 | 7.4 |
| | | | Min-$p$ | 6.8 | 7.0 | 7.6 | 7.2 | 6.4 |
| | | | Top-H | **7.4** | **7.4** | 8.0 | **7.8** | **8.4** |
| | | Prompt 3 | Top-$p$ | 7.0 | 6.4 | **8.0** | 7.6 | 8.0 |
| | | | Min-$p$ | 6.4 | 5.5 | **8.0** | 6.2 | 7.2 |
| | | | Top-H | **8.1** | **9.0** | 8.0 | **8.0** | **9.0** |
| | | Prompt 1 | Top-$p$ | 8.0 | 7.5 | **9.0** | 7.6 | 8.4 |
| | | | Min-$p$ | 7.9 | 7.8 | 8.4 | 7.5 | 8.4 |
| | | | Top-H | **8.5** | **8.3** | 8.8 | **8.4** | **9.2** |
| Phi-3-Mini | 1.5 | Prompt 2 | Top-$p$ | 7.2 | 6.8 | 7.2 | 6.6 | 7.6 |
| | | | Min-$p$ | 7.4 | 7.4 | **8.2** | **7.8** | **8.0** |
| | | | Top-H | **8.2** | **8.0** | 7.4 | 7.6 | 7.8 |
| | | Prompt 3 | Top-$p$ | 7.4 | 7.1 | **8.4** | 7.1 | 7.8 |
| | | | Min-$p$ | 6.9 | 6.6 | 7.7 | 7.2 | 7.2 |
| | | | Top-H | **8.2** | **8.0** | 8.0 | **8.0** | **8.3** |
| | | Prompt 1 | Top-$p$ | 7.2 | 7.0 | 5.6 | 6.0 | 7.0 |
| | | | Min-$p$ | **7.6** | **7.8** | **8.6** | **8.6** | 7.8 |
| | | | Top-H | **7.6** | **7.8** | **8.6** | 8.0 | **8.6** |
| | 2.0 | Prompt 2 | Top-$p$ | **8.6** | **8.8** | 5.5 | 7.0 | **8.7** |
| | | | Min-$p$ | 7.4 | 7.5 | **8.8** | 7.8 | 7.8 |
| | | | Top-H | **8.6** | 8.2 | 8.4 | **8.3** | 8.3 |
| | | Prompt 3 | Top-$p$ | **7.4** | 7.6 | 5.0 | 5.8 | 7.4 |
| | | | Min-$p$ | 6.6 | 6.6 | 8.0 | 7.0 | 7.4 |
| | | | Top-H | **7.4** | **7.8** | **8.4** | **7.6** | **8.2** |
| | | Prompt 1 | Top-$p$ | 5.0 | 4.8 | **7.2** | 5.4 | 5.2 |
| | | | Min-$p$ | 4.8 | 4.0 | 6.2 | 4.2 | 4.2 |
| | | | Top-H | **8.2** | **7.8** | **7.2** | **7.6** | **7.8** |
| | 1.0 | Prompt 2 | Top-$p$ | 7.4 | 6.8 | **8.0** | 6.8 | 7.1 |
| | | | Min-$p$ | 6.4 | 6.4 | 7.0 | 6.2 | 6.4 |
| | | | Top-H | **7.6** | **7.6** | 7.2 | **7.6** | **7.8** |
| | | Prompt 3 | Top-$p$ | 6.0 | 5.3 | **8.4** | 6.4 | 6.8 |
| | | | Min-$p$ | 6.4 | 6.0 | 7.0 | 5.8 | 6.6 |
| | | | Top-H | **7.9** | **7.6** | 8.0 | **7.3** | **8.4** |
| | | Prompt 1 | Top-$p$ | 6.0 | 5.3 | 7.8 | 5.6 | 5.6 |
| | | | Min-$p$ | 6.1 | 5.4 | 6.0 | 5.1 | 4.9 |
| | | | Top-H | **7.9** | **8.0** | **8.1** | **7.3** | **7.8** |
| Qwen2.5 | 1.5 | Prompt 2 | Top-$p$ | 7.2 | 6.8 | 7.6 | 6.8 | 7.0 |
| | | | Min-$p$ | 6.8 | 6.2 | 7.4 | 6.8 | 6.8 |
| | | | Top-H | **8.2** | **8.2** | **8.0** | **8.0** | **8.6** |
| | | Prompt 3 | Top-$p$ | 7.2 | 6.9 | 7.3 | 6.2 | 6.7 |
| | | | Min-$p$ | 6.7 | 6.5 | 7.2 | 7.1 | 7.0 |
| | | | Top-H | **7.6** | **7.5** | **7.6** | **7.2** | **8.1** |
| | | Prompt 1 | Top-$p$ | 6.0 | 5.4 | 3.8 | 3.6 | 4.4 |
| | | | Min-$p$ | 6.8 | 6.6 | 7.0 | 6.0 | 6.8 |
| | | | Top-H | **7.4** | **7.4** | **8.0** | **6.8** | **7.2** |
| | 2.0 | Prompt 2 | Top-$p$ | **7.6** | **8.0** | 4.6 | 5.8 | 7.0 |
| | | | Min-$p$ | 6.2 | 6.3 | 7.8 | 5.8 | 5.9 |
| | | | Top-H | 7.5 | 7.8 | **8.1** | **7.0** | **7.4** |
| | | Prompt 3 | Top-$p$ | **7.4** | 7.1 | 4.3 | 5.5 | 6.3 |
| | | | Min-$p$ | 6.6 | 6.5 | 5.5 | 6.4 | 6.9 |
| | | | Top-H | 7.3 | **7.4** | **8.4** | **8.2** | **8.3** |

Table 5: Evaluation metrics and the judge scores (on a scale of 1.0 to 10.0) for different LLMs, temperatures, prompts, and sampling methods. M1-M5 denote creativity, originality, narrative flow, imagery, and vitality, respectively.

| Method | Temperature = 1.0 | Temperature = 1.5 | Temperature = 2.0 |
|---|---|---|---|
| Top-p | 7.06 | 6.75 | 3.86 |
| Min-p | 7.08 | 7.11 | 6.44 |
| Top-H | **7.08** | **7.14** | **7.04** |

Table 6: MT-Bench results with LLaMA3.3-70B-Instruct.

| Method | Temperature = 1.0 | Temperature = 1.5 | Temperature = 2.0 |
|---|---|---|---|
| Top-p | 43.75 | 41.74 | 34.15 |
| Min-p | 45.76 | 42.41 | 39.29 |
| Top-H | **51.12** | **48.88** | **45.31** |

Table 7: GPQA results with LLaMA3.3-70B-Instruct.

Additionally, with the large model, we produced results with the LLM-as-judge setup described in Section 6.1.3, reported in Table 8. This demonstrates top-H's consistent improvement trend over alternatives.

| Temperature | Sampling Method | M1 | M2 | M3 | M4 | M5 |
|---|---|---|---|---|---|---|
| | Top-p | $6.05 \pm 0.24$ | $6.10 \pm 0.22$ | $8.80 \pm 0.20$ | $7.15 \pm 0.26$ | $6.95 \pm 0.22$ |
| 1.0 | Min-p | $7.05 \pm 0.22$ | $7.10 \pm 0.22$ | $\mathbf{8.85 \pm 0.20}$ | $\mathbf{7.95 \pm 0.15}$ | $8.00 \pm 0.26$ |
| | Top-H | $\mathbf{8.10 \pm 0.26}$ | $\mathbf{8.85 \pm 0.22}$ | $8.05 \pm 0.22$ | $7.60 \pm 0.20$ | $\mathbf{8.10 \pm 0.24}$ |
| | Top-p | $6.95 \pm 0.22$ | $7.80 \pm 0.20$ | $\mathbf{8.65 \pm 0.15}$ | $8.35 \pm 0.22$ | $8.40 \pm 0.20$ |
| 1.5 | Min-p | $7.10 \pm 0.30$ | $7.15 \pm 0.22$ | $8.05 \pm 0.15$ | $7.25 \pm 0.26$ | $7.15 \pm 0.22$ |
| | Top-H | $\mathbf{8.95 \pm 0.20}$ | $\mathbf{8.90 \pm 0.15}$ | $8.10 \pm 0.20$ | $\mathbf{9.00 \pm 0.22}$ | $\mathbf{8.95 \pm 0.22}$ |
| | Top-p | $7.75 \pm 0.22$ | $8.15 \pm 0.26$ | $5.55 \pm 0.22$ | $7.60 \pm 0.20$ | $7.25 \pm 0.26$ |
| 2.0 | Min-p | $8.15 \pm 0.20$ | $7.30 \pm 0.35$ | $6.25 \pm 0.22$ | $\mathbf{8.05 \pm 0.24}$ | $6.80 \pm 0.22$ |
| | Top-H | $\mathbf{8.80 \pm 0.20}$ | $\mathbf{8.20 \pm 0.30}$ | $\mathbf{7.10 \pm 0.26}$ | $8.00 \pm 0.15$ | $\mathbf{7.85 \pm 0.20}$ |

Table 8: LLM-as-judge results with LLaMA3.3-70B-Instruct.

## C.3 Human Eval

We have conducted Human Evaluation of LLM-generated texts using a setup similar to that of min-$p$, and compared with top-$p$ and min-$p$. We recruited 14 PhD students for this. We used LLaMA3.1-8B-Instruct with texts generated using a prompt adapted from the min-$p$ framework: "Write me a creative story." For each configuration, we generated three outputs, capped at 512 tokens. Participants were asked to evaluate them along quality and diversity with rating on a scale of 1–10. Results are shown in Table 9.

| Sampling Method | Creativity (T=0.7) | Coherence (T=0.7) | Creativity (T=1.0) | Coherence (T=1.0) | Creativity (T=2.0) | Coherence (T=2.0) |
|---|---|---|---|---|---|---|
| Top-p | $7.57 \pm 0.72$ | $4.78 \pm 0.93$ | $6.28 \pm 0.69$ | $5.78 \pm 0.79$ | $3.57 \pm 0.72$ | $7.57 \pm 0.62$ |
| Min-p | $6.92 \pm 0.59$ | $5.28 \pm 1.03$ | $6.21 \pm 0.77$ | $5.92 \pm 0.79$ | $6.00 \pm 0.75$ | $6.57 \pm 0.72$ |
| Top-H | $7.35 \pm 0.71$ | $5.42 \pm 0.62$ | $6.57 \pm 0.90$ | $6.42 \pm 0.91$ | $6.42 \pm 0.62$ | $7.07 \pm 0.70$ |

Table 9: Human evaluation results on creativity and coherence ratings.

## C.4 Top-H truncation threshold

In this section, we demonstrate how top-H addresses the limitations of min-$p$ sampling, as illustrated in Fig. 1, which served as our motivational case study. Comparing the two distributions, we observe that distribution A exhibits greater randomness, with a higher proportion of low-probability tokens relative to distribution B. This observation is supported by their entropy values: distribution A has an entropy of 4.28, while distribution B has a lower entropy of 3.71. Consequently, an optimal

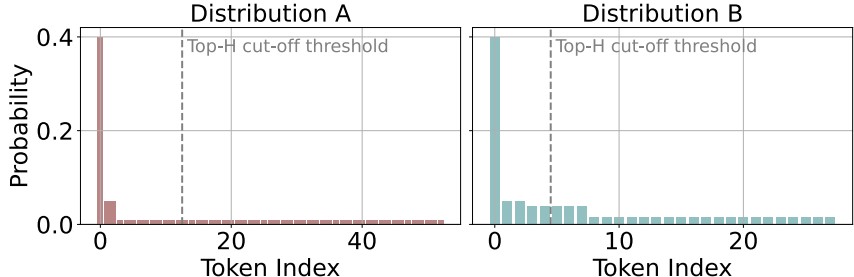

Figure 6: Probability distribution of two different types with associated top-H thresholds.

decoding strategy would be expected to allocate a larger sampling pool in scenario A, reflecting the model's lower confidence. This is precisely how the top-H decoding method operates. When applied with $\alpha = 0.4$, the resulting entropy thresholds are shown in Fig. 6. As illustrated, top-H assigns a significantly larger token set to distribution A to accommodate its higher uncertainty—an adjustment that min-$p$ fails to make. Moreover, in scenario B, top-H retains several high-probability tokens that min-$p$ erroneously excludes. Therefore, top-H effectively addresses both key shortcomings of min-$p$ sampling in such settings.

## C.5 Impact of the $\alpha$ Parameter

Table 10 reports GPQA accuracy and the average sampling pool size across different $\alpha$ values. The experiment is done using the `LLaMA3.1-8B-Instruct` model with temperature $T = 1.5$. These results show that:

1. Larger $\alpha$ values slightly reduce accuracy, which aligns with the nature of GPQA's graduate-level questions that benefit from more confident (less diverse) answers.

2. Sampling pool size increases with $\alpha$, providing more generative options and supporting the creativity aspect observed in Figure 4.

| $\alpha$ | 0.10 | 0.15 | 0.20 | 0.25 | 0.30 | 0.35 | 0.40 | 0.45 | 0.50 | 0.55 | 0.60 | 0.65 | 0.70 | 0.75 | 0.80 | 0.85 | 0.90 |
|---|---|---|---|---|---|---|---|---|---|---|---|---|---|---|---|---|---|
| Accuracy on GPQA | 0.3085 | 0.3085 | 0.3095 | 0.3085 | 0.3125 | 0.3103 | 0.3058 | 0.3050 | 0.2869 | 0.2937 | 0.2879 | 0.2790 | 0.2655 | 0.2879 | 0.2612 | 0.2656 | 0.2701 |
| Average sampling pool size | 1.01 | 1.03 | 1.20 | 1.48 | 1.90 | 2.53 | 3.48 | 4.74 | 6.94 | 9.11 | 11.79 | 15.77 | 21.35 | 27.99 | 36.28 | 47.06 | 59.28 |

Table 10: GPQA accuracy and average sampling pool size across different $\alpha$

## C.6 Comparison of top-H to Mirostat method

In addition to the results obtained with $\eta$-sampling, we include further comparisons with Mirostat (Basu et al., 2020) Table 11 for MTBench and Table 12 for GPQA, serving as an additional entropy-aware baseline. Unless otherwise specified, all decoding and evaluation configurations follow those in the paper. For Mirostat, we set the target entropy parameter to $\tau = 3$.

| Temperature | LLaMA3.1-8B-Instruct | | Phi-3-Mini | | Qwen2.5 3B | |
|---|---|---|---|---|---|---|
| | Top-H | Mirostat | Top-H | Mirostat | Top-H | Mirostat |
| 1.0 | **6.788** | 6.375 | **6.819** | 6.600 | **5.956** | 5.369 |
| 1.5 | **6.819** | 5.594 | **6.556** | 5.500 | **5.513** | 4.469 |
| 2.0 | **6.438** | 5.519 | **6.056** | 5.269 | **4.519** | 4.256 |

Table 11: MTBench results comparing Top-H and Mirostat. Top-H wins in all 9 settings. Averaged over all models and temperatures, Top-H achieves 6.163 vs. Mirostat 5.439 (+0.724 absolute, +13.3%).

# D Limitations

In this paper, we introduced a novel sampling method—top-H—as a greedy solution to the NP-hard *entropy-constrained mass maximization* (ECMM) problem. While top-H demonstrates strong

| Temperature | LLaMA3.1-8B-Instruct | | Phi-3-Mini | | Qwen2.5 3B | |
|---|---|---|---|---|---|---|
| | Top-H | Mirostat | Top-H | Mirostat | Top-H | Mirostat |
| 1.0 | 29.24 | **30.36** | **32.37** | 30.13 | **28.79** | 28.35 |
| 1.5 | **30.58** | 25.67 | **30.80** | 29.02 | **27.90** | 26.34 |
| 2.0 | **28.79** | 28.79 | **30.80** | 29.91 | **28.12** | 27.79 |

Table 12: GPQA results comparing Top-H and Mirostat. Top-H outperforms Mirostat in 7 of 9 settings, with one Mirostat win at LLaMA-8B (T = 1.0). Averaged over all models and temperatures, Top-H achieves 29.71 vs. Mirostat 28.48 (+1.23 absolute, +4.3%).

empirical performance, it does not provide general competitive guarantees that apply across a broad range of distributions. Moreover, the hyperparameter $\alpha$ was tuned manually, even though the method exhibits robustness to its variation. Designing an algorithm that offers a provable approximation ratio and can dynamically adapt the entropy threshold $\alpha$ remains an important direction for future work.

# E Broader Impact

Top-H sampling enhances the coherence and creativity of text generated by large language models, especially at high temperatures. This can positively impact applications such as creative writing, education, and human-AI interaction by making outputs more diverse and engaging. Its efficiency and ease of integration also support broader accessibility in open-source settings. However, the same improvements in fluency could be misused to generate more persuasive disinformation or evade content moderation. While top-H is a general-purpose sampling method, we recommend pairing it with safety mechanisms and monitoring in sensitive deployments. Open-sourcing our implementation and providing clear usage guidelines will support responsible adoption and further research.

