# OpenReview forum: "Top-H Decoding: Adapting the Creativity and Coherence with Bounded Entropy in Text Generation"
_NeurIPS.cc/2025/Conference — NeurIPS 2025 poster_

### Official Review · Reviewer_bsFC · 2025-06-29

**Clarity:** 3
**Significance:** 2
**Originality:** 3
**Rating:** 4
**Confidence:** 4

**Summary:**

This paper presents top-H decoding, a truncated sampling strategy that aims to balance creativity and coherence.
The paper first identifies the limitations in existing truncation-based sampling methods like min-p sampling, which relies solely on the top token probability as a confidence heuristic.
They formulate the problem as an entropy-constrained minimum divergence (ECMD) optimization, prove its equivalence to an entropy-constrained mass maximization (ECMM) problem, and show that ECMM is NP-hard. Top-H provides a greedy approximation that dynamically selects tokens to bound the entropy of the truncated distribution while minimizing divergence from the original distribution.
Experimental results show that top-H Decoding is competitive across creative writing and reasoning tasks on various benchmarks.

**Questions:**

Please refer to the weakness section

**Ethical Concerns:**

["NO or VERY MINOR ethics concerns only"]

**Final Justification:**

The authors have addressed my concerns

**Limitations:**

Yes

**Quality:**

3

**Strengths And Weaknesses:**

**Strengths:**

- The paper provides a rigorous mathematical framework by formulating the creativity-coherence trade-off as an optimization problem. The mathematical proofs are well-presented.

- The paper conducts extensive experiments across diverse benchmarks, including creative writing, reasoning tasks (GSM8K, GPQA), and LLM-as-a-judge evaluations to demonstrate the efficacy of the method.

- The proposed method is computationally efficient and can be applied easily.

**Weaknesses:**

- While the paper proves NP-hardness and provides a greedy solution, it lacks formal approximation bounds for the proposed algorithm. The empirical evaluation in Figure 5 suggests good performance relative to the optimal solution, but without theoretical guarantees, the worst-case behavior remains unclear. This is a significant limitation for a method with strong theoretical motivations.
The algorithm doesn't consider alternative orderings or lookahead strategies. For instance, temporarily skipping a token that causes large entropy increases might allow inclusion of multiple smaller probability tokens that together contribute more mass while better respecting the entropy constraint.

- The paper positioned itself as an entropy-aware approach, but the experiments are more about comparing it with sampling-based methods, given that only the $\eta$-sampling is presented for entropy approaches. More comprehensive comparisons with existing entropy-based methods would provide better context for the contributions.

- While the paper claims the algorithm is computationally efficient, there is insufficient analysis of the actual computational cost compared to simpler baselines. The entropy calculation and iterative token selection process likely introduces overhead that should be quantified.

- The paper only provides quantitative evaluations of the results. Further qualitative studies (e.g., human evaluations) are expected to demonstrate the balance of randomness and coherence of the generated content.

---

> ### Author Rebuttal · Authors · 2025-07-31
>
> We thank the reviewer  bsFC for the insightful feedback and clarify their concerns below.
>
> >**Discussion on formal approximation bound for the Top-H**
>
> While the primary objective of our work was to empirically demonstrate the efficacy of the top-H decoding, we largely intend to keep the detailed theoretical underpinning of the error bound analysis as a part of future research. Nevertheless, we now attempt to demonstrate a worst-case bound for top-H under the specific assumption of a probability distribution.
>
> **Zipf model.** Fix a vocabulary of size $n$ and exponent $s>1$. We assume the sorted next-token probabilities obey the classical Zipf / regularly varying law:
>
> $
> p_i := \frac{i^{-s}}{H_{n,s}},\qquad
> H_{n,s}=\sum_{j=1}^{n}j^{-s}.
> \quad (1)
> $
>
> Empirically, language-model logits are well approximated by $s\in[1.05,1.20]$ [1]; hence the assumption captures current practice.
>
> **Why a Zipf assumption matters.**
> Because ECMM is NP‑hard (Appendix C), exact polynomial‑time solutions are unlikely. In the absence of structural assumptions, a constant-factor approximation guarantee for ECMM is highly unlikely unless P=NP, as our NP-hardness proof relies on a gap-preserving reduction from Cardinality-Constrained Subset Sum. This structure is known to be fundamentally difficult to approximate, a standard result in computational complexity theory [2]. However, LLM logits consistently follow heavy‑tailed (Zipf / regularly‑varying) laws, so analysing this regime yields practically relevant bounds.
>
> **Notation.** We write $M(k)=\sum_{i\le k}p_i$ for the prefix mass, $T(k)=1-M(k)$ for the tail mass, and $H(k)$ for the entropy of the normalized prefix distribution $q^{(k)}_i=p_i/M(k)$.
>
> ---
>
> ### Preliminaries
>
> **Lemma 1 (Monotonicity of prefix entropy).**
> For $1 \leq k < n$, $H(k) < H(k+1)$.
>
> ***Proof.***
> Please refer to the proof for Theorem 3 of the main manuscript.
>
> **Lemma 2 (Tail mass bound).**
> For $k<n$,
>
> $\frac{(k+1)^{1-s}-n^{1-s}}{(s-1)H_{n,s}} \le T_{n}(k)\le\frac{k^{1-s}}{(s-1)H_{n,s}}.$
>
> When $n\gg k$, the numerator difference is $o(k^{1-s})$, so the upper bound is asymptotically tight.
>
> ***Proof.***
> Apply upper and lower Riemann sums to $\int_{k}^{n}x^{-s}\,dx$.
>
> **Lemma 3 (Prefix entropy asymptotics).**
> There exist constants $c_s,C_s>0$ such that
>
> $c_s+\frac{s}{s-1}\log\left(\frac{k}{2}\right)\le H(k)\le C_s+\frac{s}{s-1}\log k \quad \text{for all } k\ge 2.$
>
> ***Proof.***
> Combine Lemma 2 with integral bounds for $\sum i^{-s}\log i$.
>
> ---
>
> ### Depth of the greedy prefix
>
> Let $k_g:=\max\{k:H(k)\le \alpha H(n)\}$. Lemma 1 implies $k_g$ is well defined.
>
> **Lemma 4 (Growth rate of $k_g$).**
> There exist constants $a_s,b_s>0$ such that $a_s n^{\alpha}\le k_g\le b_s n^{\alpha}$.
>
> ***Proof.***
> Insert Lemma 3 into the defining inequality and solve for $k_g$.
>
> ---
>
> ### Mass captured by **Top-H**
>
> Write $\Gamma_g=M(k_g)$. This is the mass captured by the greedy solution. By construction, this solution is valid as it satisfies the entropy constraint $H(k_g) \le \alpha H(n)$.
>
> ### Tight upper bound for ECMM
>
> Let $S^\star$ be *any* subset satisfying the entropy constraint, and set $\Gamma_\star=\sum_{i\in S^\star}p_i$. We want to bound the maximum possible value of $\Gamma_\star$.
>
> The greedy algorithm selects the prefix $[k_g]$ and captures a mass of $\Gamma_g = M(k_g) = 1 - T(k_g)$. The total mass of all tokens *not* in the greedy solution is, by definition, the tail mass $T(k_g)$.
>
> Any other valid solution $S^\star$ can, at best, capture the mass of the greedy solution *plus* some portion of the remaining tail mass. The absolute maximum mass any solution can capture is 1 (the entire vocabulary). Therefore, the maximum possible improvement any optimal solution $\Gamma_\star$ can have over the greedy solution $\Gamma_g$ is bounded by the tail mass that the greedy algorithm left behind:
>
> $\Gamma_\star - \Gamma_g \le 1 - \Gamma_g = 1 - M(k_g) = T(k_g). \quad (\dagger)$
>
> This gives us a direct upper bound on the additive gap between the optimal and greedy solutions.
>
> ---
>
> **Theorem 1 (Distribution-dependent additive guarantee).**
> Under equation (1), for every $n\ge 4$,
>
> $\Gamma_\star - \Gamma_g \le T_{n}(k_g) \le \frac{k_g^{1-s}}{(s-1)H_{n,s}} = \mathcal{O}(n^{-\alpha(s-1)}).$
>
> ***Proof.***
> From ($\dagger$) we have the additive gap $\Gamma_\star-\Gamma_g\le T(k_g)=T_{n}(k_g)$. Lemma 2 bounds $T_{n}(k_g)$ by $\frac{k_g^{1-s}}{(s-1)H_{n,s}}$. Finally, Lemma 4 gives $k_g=\Theta(n^{\alpha})$, so the gap decays as $\mathcal{O}(n^{-\alpha(s-1)})$.
>
>
> >**Discussion on the effectiveness of greedy:**
>
> We understand that the approach of dropping high-probability tokens could in principle beat the greedy prefix by allowing more low-mass tokens from the tail. However, as discussed in the paper, this problem (ECMM) is NP hard, and we approximate the solution via a practically feasible greedy approach. Notably, we empirically demonstrate the effectiveness of this greedy based top-H approach to be superior to the existing SoTA.
> Additionally, under the constrained distribution of the Zipfian regime, the greedy prefix is constructed to generally maximize mass under minimal entropy growth. Because entropy increases monotonically with each added token, and the most probable tokens usually contribute less to entropy per unit mass, the greedy approach reaches the entropy threshold more efficiently than any alternative.
>
>
>
>
>
>
> >**Comparison of top-H to other entropy-based sampling method:**
>
> The manuscript has results with $\eta$-sampling. We now present more comparison with
> Mirostat [3] on MTBench (R3T1), and GPQA (R3T2) as an additional entropy‑aware baseline. Unless otherwise noted, all decoding/eval settings match the main paper; we set $\tau$ = 3 for Mirostat.
>
> ### R3T1
> |                | **LLaMA-8B** | **LLaMA-8B**| **Phi-3**    | **Phi-3**  | **Qwen-2.5 3B** |**Qwen-2.5 3B**  |
> |:-:|:-:|:-:|:-:|:-:|:-:|:-:|
> |**Temp** | Top-H               | Mirostat          | Top-H        | Mirostat    | Top-H                 | Mirostat|
> | 1.0         | **6.788**          | 6.375              | **6.819**   | 6.600        | **5.956**            |5.369  |
> | 1.5         | **6.819**          | 5.594              | **6.556**   | 5.500        |**5.513**             |4.469  |
> | 2.0         | **6.438**          | 5.519              | **6.056**   | 5.269        | **4.519**            |4.256 |
>
> Top‑H wins in 9/9 settings. Averaged over all models and temperatures, Top‑H = 6.163 vs. Mirostat = 5.439 (+0.724 absolute, +13.3%).
>
>
> ### R3T2
>
> |                | **LLaMA-8B** | **LLaMA-8B**| **Phi-3**    | **Phi-3**  | **Qwen-2.5 3B** |**Qwen-2.5 3B**  |
> |:-:|:-:|:-:|:-:|:-:|:-:|:-:|
> |**Temp** | Top-H               | Mirostat          | Top-H        | Mirostat    | Top-H                 | Mirostat|
> | 1.0         | 29.24                | **30.36**        | **32.37**   | 30.13        | **28.79**            |28.35  |
> | 1.5         | **30.58**          | 25.67              | **30.80**   | 29.02        |**27.90**             |26.34  |
> | 2.0         | **28.79**          | 28.79              | **30.80**   | 29.91        | **28.12**            |27.79  |
>
> GPQA: Top‑H wins 7/9 settings (1 Mirostat win at LLaMA‑8B, T=1.0). Averages: Top‑H = 29.71 vs. Mirostat = 28.48 (+1.23 absolute, +4.3%).
>
> >**Computational cost of top-H**
> Please refer to the table  R1T1 and corresponding response to reviewer ELnF .
>
>
>  >**Human Evaluation results**
> Please refer to the table R1T2 and corresponding response to reviewer ELnF .
>
> ---
> ## References
> [1] Gerlach, Martin, and Eduardo G. Altmann. "Stochastic model for the vocabulary growth in natural languages." Physical Review X 3.2 (2013): 021006.
>
> [2] Papadimitriou, Christos H. "Computational complexity." Encyclopedia of computer science. 2003. 260-265.
>
> [3] Mirostat: A neural text decoding algorithm that directly controls perplexity, arXiv.

---

> > ### Comment · Reviewer_bsFC · 2025-08-05
> >
> > Thank you for the clarification. Most of my concerns are well-addressed, and I will keep my scores.

---

> > > ### Author Response · Authors · 2025-08-05
> > > **Re: Thanks**
> > >
> > > We thank the reviewer **bsFC** for their kind acknowledgment, and are glad to note that our rebuttal addressed the pressing concerns!
> > >
> > > We will be happy to incorporate the rebuttal results to the main draft, to add more clarity to the contributions.
> > >
> > > Regards
> > >
> > > Authors

---

> ### Author Response · Authors · 2025-08-04
>
> Dear Reviewer,
>
> Thank you for the time and effort you’ve put into reviewing our submission. We’ve carefully considered each of your comments and concerns, and have done our best to address them thoroughly in our rebuttal with the necessary theoretical or empirical evidences.
>
> We understand that time-bound reviewing process is demanding. However,  as the deadline of discussion is approaching soon, we would greatly appreciate it if you could take a moment to revisit our response and, if possible, review the scores. With the enhanced thoroughness driven by your feedback, we believe top-H decoding has emerged as a promising alternative to the existing approaches.
>
> Your feedback is invaluable and can help ensure that our clarifications are appropriately considered in the final decision.
>
> Please don’t hesitate to reach us out in case any further discussion is needed.
>
> Best regards,
>
> Authors

---

### Official Review · Reviewer_2JSm · 2025-07-04

**Clarity:** 2
**Significance:** 3
**Originality:** 2
**Rating:** 3
**Confidence:** 3

**Summary:**

The paper introduces Top-H, a truncated LLM sampling approach, which greedily selects the most-confident tokens up to a fraction of the original entropy for sampling. Empirical evaluation using various models (LLaMA3.1, Qwen2.5, Phi-3) on several tasks (creative writing, reasoning, QA, that is on Alpha-Eval & MT-bench, GSM8K CoT, and GPQA) shows better performance vs baselines such as top-p, top-k, min-p, and eta-sampling.

**Questions:**

1. **Fundamental theoretical validity**: Given that reasonable truncated sampling should never drop higher-confidence tokens for lower-confidence ones, why formulate ECMD to allow such counterintuitive solutions? Isn't the "greedy approximation" actually the correct algorithm, making the NP-hardness analysis irrelevant? Can you construct a case, where we would want to drop the maximum confidence token, or generally tokens with higher confidence in favor of tokens with lower confidence?
2. **Alternative theoretical foundation**: Can you reformulate the approach based on maximum entropy principles over confidence-ordered contiguous token sets? This seems like a more natural and theoretically sound foundation than the current ECMD formulation.
3. **Statistical significance**: Can you provide confidence intervals or statistical significance tests for your results? The current reporting of only means over 5 runs is insufficient to establish that improvements are statistically meaningful.
4. **Scalability validation**: How does top-H perform on larger models (70B+) where baseline methods have been tested? The current evaluation on only small models limits the generalizability of claims.
5. **Judge reliability**: Given the heavy reliance on GPT-4o evaluation, can you validate judge reliability through inter-annotator agreement with human evaluators or comparison with multiple judge models?

**Ethical Concerns:**

["NO or VERY MINOR ethics concerns only"]

**Final Justification:**

The human eval has flaws (sadly, so does min-p in my opinion). But this one has even fewer participants (14).

I'm also still not convinced by the necessity for all the theoretical complexity when the same greedy algorithm can be explained and motivated differently as well, following existing literature. There is no need to field NP-hard problems and then simplify them to a greedy algorithm in line with other popular sampling methods. The authors have agreed "that a maximum‑entropy (ME) view over confidence‑ordered prefixes is intuitive and top‑H can be stated that way". In my opinion, the simpler motivations should win, and the current theory-heavy setup distracts from the actual algorithm and its empirical examination.

There is also another discrepancy in the appendix D in the given implementation for top-H that also seems to use top-K by default, which has not been mentioned anywhere else in the paper.

Overall, I think keeping my current score is warranted.

**Limitations:**

yes

**Quality:**

2

**Strengths And Weaknesses:**

**Strengths:**

1. **Comprehensive empirical evaluation**: The experiments span multiple small state-of-the-art models (\<=8B params), diverse task types, and include both automatic metrics and LLM-as-judge evaluations. The results show improvements, especially in high-temperature settings..
2. **Clear motivation and practical relevance**: The motivational example conceptually shows limitations of existing methods, i.e. min-p sampling, and the proposed solution integrates seamlessly into existing generation pipelines.
3. **Well-written and accessible**: The paper is clearly structured with good intuitive explanations alongside technical details.

**Weaknesses:**

1. **Inadequate statistical reporting**: The paper reports only means over 5 runs without confidence intervals, making it impossible to assess statistical significance of the reported improvements.
2. **Limited experimental scope**: Only small models are evaluated (largest is 8B), while e.g. min-p  baseline studies, whose evaluations this paper seems to copy, included Llama 70B. No human evaluation is conducted, and heavy reliance on GPT-4o as judge introduces potential systematic biases without validation of judge reliability.
3. **Fundamental theoretical flaws**: The core theoretical contribution is undermined by several critical issues:
   * **Misguided problem formulation**: Any reasonable truncated sampling should never drop a token with higher confidence in favor of one with lower confidence. This principle immediately constrains truncated sampling to contiguous sets of tokens sorted by confidence.
   * **Theory contradicts intuition**: If the optimal solution to ECMD discards high-confidence tokens in favor of lower-confidence ones, then ECMD is the wrong objective function. The greedy algorithm isn't approximating an NP-hard problem: it's actually solving the correct problem that ECMD fails to capture (otherwise it wouldn’t be NP-hard).
   * **Redundant complexity**: Top-H follows naturally from the principle of maximum entropy within confidence-ordered contiguous sets (maximum uninformativeness principle) for a given dynamic entropy threshold, rendering the complex ECMD formulation unnecessary.
   * **Unjustified design choices**: The choice of Jensen-Shannon divergence lacks strong theoretical motivation (and the α parameter appears empirically driven rather than principled).
4. **Vacuous theoretical results**: Theorem 3 (termination guarantee) is essentially meaningless since any algorithm operating on finite vocabulary with per-token decisions is trivially guaranteed to terminate. The result provides no useful insight.
5. **Exaggerated claim(s)**: "solving the ECMD problem can ideally balance creativity and coherence" is an unnecessarily strong and broad claim that is not evidenced by the paper.

---

> ### Author Rebuttal · Authors · 2025-07-29
>
> We thank the reviewer  2JSm for the insightful feedback and clarify their concerns below.
>
> > **Inadequate statistical reporting:**
>
> We have performed 10 runs and computed standard deviation across all runs for each config. We see the max std dev to be $\pm 0.35$; we commit to update Table 3 with the detailed results.
>
> >**Validation with large model:**
>
> We replicated the MT-Bench (R2T1) and GPQA (R2T2) experiments using **LLaMA3.3-70B-Instruct** with the setup of section 6.1 in the paper. Specifically, top-H outperforms min-p by up to 6.0% and 6.47% on MT-Bench  and GPQA, respectively.
> ### R2T1
> | Method  | T = 1.0 | T = 1.5 | T = 2.0 |
> |:--:|:--:|:---:|:--:|
> | top-p   | 7.06        |  6.75       |  3.86       |
> | min-p   |      7.08   |   7.11      |   6.44      |
> | top-H   |   **7.08**      |    **7.14**     |    **7.04**     |
>
> ### R2T2
> | Method  | T = 1.0 | T = 1.5 | T = 2.0 |
> |:--:|:--:|:--:|:--:|
> | top-p   | 43.75        |  41.74      |  34.15       |
> | min-p   |      45.76   |   42.41      |   39.29      |
> | top-H   |   **51.12**      |    **48.88**     |    **45.31**     |
>
> Additionally, with the large model, we produced results with LLM-as-judge setup described in Section 6.1.3, and is reported in Table R2T3. This demonstrates top-H’s consistent improvement trend over alternatives.
>
> ### R2T3
>
> | Temperature | Method |  $~~~~~~$ M1   | $~~~~~~$ M2   |  $~~~~~~$ M3   | $~~~~~~$ M4   | $~~~~~~$ M5   |
> |:-----------:|:----------------:|:------:|:------:|:------:|:------:|:------:|
> | T = 1.0     | top-p           | 6.05 $\pm$ 0.24 | 6.10 $\pm$ 0.22 | 8.80 $\pm$ 0.20 | 7.15 $\pm$ 0.26 | 6.95 $\pm$ 0.22 |
> |             | min-p           | 7.05 $\pm$ 0.22 | 7.10 $\pm$ 0.22 | **8.85** $\pm$ 0.20 | **7.95** $\pm$ 0.15 | 8.00 $\pm$ 0.26 |
> |             | top-h           | **8.10** $\pm$ 0.26 | **8.85** $\pm$ 0.22 | 8.05 $\pm$ 0.22 | 7.60 $\pm$ 0.20 | **8.90** $\pm$ 0.20 |
> | T = 1.5     | top-p           | 6.95 $\pm$ 0.22 | 7.80 $\pm$ 0.2 | **8.65** $\pm$ 0.15 | 8.35 $\pm$ 0.22 | 8.40 $\pm$ 0.20 |
> |             | min-p           | 7.10 $\pm$ 0.30 | 7.15 $\pm$ 0.22 | 8.05 $\pm$ 0.15 | 7.25 $\pm$ 0.26 | 7.15 $\pm$ 0.22 |
> |             | top-h           | **8.95** $\pm$ 0.20 | **8.90** $\pm$ 0.15 | 8.10 $\pm$ 0.2 | **9.00** $\pm$ 0.22 | **8.95** $\pm$ 0.22 |
> | T = 2.0     | top-p           | 7.75 $\pm$ 0.22 | 8.15 $\pm$ 0.26 | 5.55 $\pm$ 0.22 | 7.60 $\pm$ 0.20 | 7.25 $\pm$ 0.26 |
> |             | min-p           | 8.15 $\pm$ 0.20 | 7.30 $\pm$ 0.35 | 6.25 $\pm$ 0.22 | 8.05 $\pm$ 0.24 | 6.80 $\pm$ 0.22 |
> |             | top-h           | **8.80** $\pm$ 0.20 | **8.20** $\pm$ 0.30 | **7.10** $\pm$ 0.26 | **8.00** $\pm$ 0.15 | **7.85** $\pm$ 0.20 |
>
>
> >**Human evaluation results:**
>
> Please refer to the table R1T2 of the rebuttal response for reviewer ELnF.
>
> >**On misguided problem formulation-explanation of greedy solution to not be optimal:**
>
> While the reviewer’s intuition largely holds in practice, there are certain situations where the **greedy solution wouldn’t be optimal** and there are some nuances only ECMM captures. We explain it below.
>
> *Case for a lower-p token might be preferred over a higher one:* Entropy responds in a highly non-linear way after renormalization. When the entropy cap is tight and we have multiple high-probability (dominant) tokens, the distribution can be so flat that adding *any* extra token violates $H(q)\\le \\alpha H(p)$. Dropping one of the dominant tokens sharpens $q$, letting us include several medium-probability tokens whose **combined** mass exceeds what was lost, thereby increasing $\\Gamma\_S$. We present an example of such in sorted order, resembling an hypothetical set of probabilities generated by an LLM.
> ```
> [4.438e-01, 4.006e-01, 1.027e-01, 3.320e-02,
>  8.733e-03, 5.801e-03, 2.413e-03, 1.468e-03,
>  4.460e-04, 4.286e-04, 1.804e-04, 5.383e-06,
>  5.205e-07, 3.760e-08, 1.521e-12]
> ```
> Let the entropy threshold with $\alpha=0.4$ be $0.6796$. The greedy solution takes only the first token with a probability of 4.438e-01 and stops, due to threshold exceeding issue. However, the optimal solution $\\Gamma\_S$ would be subset `[4.438e-01, 1.027e-01, 3.320e-02, 1.804e-04, 5.383e-06, 5.2056e-07, 3.760e-08, 1.521e-12]` with an entropy of 0.6777 and a retained mass of 0.580 \> 0.4438 (greedy solution). Thus, there are special cases where the optimal solution to ECMM would prefer a token with a lower probability over a token with a higher probability.
>
> >**Exaggerated claim(s): "solving the ECMD problem can ideally balance creativity and coherence"**
>
> Based on the above example, we argue that *solving the ECMM problem can yield solutions better than the greedy algorithm, particularly for corner cases.* This rationale also helps ECMM maintain a significantly better creativity-coherence trade-off over the SoTA: maximizing $\\Gamma\_S$ enforces coherence, while the entropy cap gates creativity and prevents coherence degradation. Top-H implements this balance greedily: it locks in the most probable tokens first, then expands until the entropy budget is about to be exceeded.
>
> >**The Choice of JSD:**
>
> We chose JSD because it is a standard, bounded, and symmetric measure of divergence. Crucially, as we prove in Theorem 1, minimizing the JSD between our distributions is mathematically equivalent to maximizing the retained probability mass ($\\Gamma\_S$​). This is a powerful and intuitive result: our theoretically grounded objective of minimizing divergence simplifies to the very practical goal of keeping the most probable tokens, which directly addresses the coherence objective.
>
> >**Vacuous theoretical results (Theorem 3):**
>
> We agree that *mere* termination on a finite vocabulary is trivial. Theorem 3 is not about finiteness, rather establishes a **strict monotonicity property of entropy under probability‑ordered inclusion**. Thus, there is a single crossing point of the constraint $\(H(q_k)\le \alpha H(p)\)$ that greedy algorithm finds. This ensures: 1. Strict monotonicity guaranteeing a *unique* stopping index. Without it, we could overshoot/undershoot the entropy cap. 2. The theorem is the **formal link** between the entropy constraint and the greedy prefix growth, making it an inevitable consequence—not just a heuristic. 3. The ordering assumption is what turns an NP‑hard subset problem into a linear‑time scan.
>
> >**Alternative theoretical foundation:**
>
> We agree that a maximum‑entropy (ME) view over confidence‑ordered prefixes is intuitive and top‑H can be stated that way. However, ECMM formulation is strictly more general (it does **not** assume contiguity), yields the divergence–mass equivalence (Thm. 1), and provides the optimality/complexity guarantees that the ME‑prefix view alone cannot.
>
> #### 1. Maximum-Entropy restatement of Top-H (under prefix restriction)
>
> Let $p$ be sorted by probability: $p_{(1)} \ge p_{(2)} \ge \dots$.
> For the prefix $S_k = \{1,\dots,k\}$, define
> $q_k(i) = \frac{p_{(i)}}{\sum_{j \le k} p_{(j)}} \quad (i \le k).$
>
> If we **restrict ourselves to such prefixes**, then choosing “the most uninformed (highest entropy) distribution that still respects the entropy cap” reduces to:
>
> $k^\star = \max\{k \mid H(q_k) \le \alpha H(p)\}.$
>
> Since $H(q_k)$ increases monotonically with $k$ when tokens are added in descending $p_{(i)}$ (Theorem 3), this rule is identical to Top-H that adding tokens until the entropy cap would be violated. Thus, under prefix restriction, the top-H is equivalent to the maximum-entropy set (subject to the $\alpha H(p)$ threshold) over the confidence-ordered prefix.
>
> #### 2. Relation to ECMD/ECMM
>
> ECMD objective starts from minimizing divergence of $p$ (fidelity) while bounding randomness. We show that minimizing JSD under the same entropy cap is equivalent to maximizing retained probability mass $\Gamma_S = \sum_{i \in S} p_i$ (ECMM). Over ordered prefixes, $\Gamma_{S_k}$ is monotone in $k$, so the ECMM optimum picks the same $k^\star$.
>
> Thus, the ME-prefix view *rephrases* the same greedy rule—but only after imposing the prefix constraint.
>
> >**Why retain ECMD as the main formulation?**
>
> 1. *No prefix assumption:* ECMD does not assume contiguity. The fact that greedy prefixes are (near-)optimal is **derived**, not assumed.
> 2. *Theoretical guarantees:* ECMD/ECMM yields (i) the divergence–mass equivalence (Theorem. 1) and (ii) NP‑hardness of the unrestricted problem—explaining why a greedy approximation is appropriate.
> 3. *Interpretability & control:* The entropy ratio $\(\alpha\)$ gives a clear knob linking creativity (entropy) to fidelity (mass), independent of the prefix framing.
> 4. *Unification:* The ECMD lens naturally relates Top‑H to other decoding schemes (temperature, nucleus) within one objective-driven framework.

---

> > ### Comment · Reviewer_2JSm · 2025-08-04
> >
> > Thank you for the response!
> >
> > The results for the bigger model are good to see. I'm satisfied with this.
> >
> > Sadly, the human eval results (like those in min-p) are flawed/not convincing. Sampling 3 outputs per method and keeping those fixed while evaluating them using multiple raters does not capture diversity (and the other properties) well at all. Having multiple raters rate those three fixed outputs only reduces the variance of a point estimate but fails to capture anything meaningful about the distribution, in particular at higher temperatures. (3 samples for creativity etc. is very little, or similarly a single triplet for diversity.)
> >
> > Thank you for the example re ECMM and greedy sampling. I agree that ECMM can lead to cases where the top confidence tokens are categorically dropped. However, the question remains: when is that good? When sampling locally, potentially dropping the most confident tokens sounds like a bad idea, as it would likely harm performance. As such, I still question why the ECMM formulation is needed at all, especially given that the actual method you propose outperforms all other methods and does not utilize this formulation, instead relying on a simple greedy method that can be easily explained differently.
> >
> > As such, I am inclined to keep my current score because I would rather not have a paper with so much unnecessary theoretical complexity accepted that could easily be explained differently.
> >
> > Finally, I have just noticed that the proposed algorithm in §D includes a top-n=100 parameter, so it does not apply the entropy constraint to all token candidates but only subselects from the top-n tokens. This is not mentioned elsewhere in the paper. What was used for the experiments?

---

> ### Author Response · Authors · 2025-08-04
>
> Dear Reviewer,
>
> Thank you for the time and effort you’ve put into reviewing our submission. We’ve carefully considered each of your comments and concerns, and have done our best to address them thoroughly in our rebuttal with the necessary theoretical or empirical evidences.
>
> We understand that time-bound reviewing process is demanding. However,  as the deadline of discussion is approaching soon, we would greatly appreciate it if you could take a moment to revisit our response and, if possible, review the scores. With the enhanced thoroughness driven by your feedback, we believe top-H decoding has emerged as a promising alternative to the existing approaches.
>
> Your feedback is invaluable and can help ensure that our clarifications are appropriately considered in the final decision.
>
> Please don’t hesitate to reach us out in case any further discussion is needed.
>
> Best regards,
>
> Authors

---

> ### Author Response · Authors · 2025-08-05
> **Official comment to Reviewer 2JSm (part 1/2)**
>
> We thank the reviewer 2JSm for their response to the rebuttal, and happy to note that some parts of the response were convincing. We now further clarify on the remaining.
>
> >**Clarity to the Human eval setup**:
>
> We would like to clarify the rationale behind our human evaluation setup. Our design closely follows the _min-p_ paper (accepted as ICLR 2025 Oral). This choice was made, primarily to ensure a fair and direct comparison with _min-p_, which represents the current state of the art. Due to the limited time available during the rebuttal phase, we restricted the number of outputs per configuration to three. Increasing this number would have significantly increased the burden on participants and was not feasible within the limited timeline. However, we are committed to conducting a more comprehensive human evaluation with a larger set of samples for the camera-ready version of the paper, should it be accepted.
>
>
> >**Clarification on when is "dropping high confidence tokens is good"?**
>
> As we highlighted in the earlier responses, despite it may seem counterintuitive, dropping the highest-probability token can **improve overall generation quality** in certain settings. This happens when we care more about **maintaining diversity and coherence across multiple plausible options** rather than simply following the single most likely path. Following are few examples:
>
> a) Open-ended generation (e.g., storytelling, ideation): The model faces multiple valid directions for the next word—like the first sentence of a story or brainstorming ideas. The top token may dominate due to a small margin, but **excluding it allows several alternative directions** to enter the beam, giving the decoder a richer set of continuations.
>
> b) Multi-modal or ambiguous contexts: The probability mass is spread across **distinct semantic modes** (e.g., “cat”, “dog”, “rabbit” are all plausible continuations). Greedy will lock into just one mode (e.g., “cat”), suppressing others. ECMM allows each cluster to be represented by a token.
>
> c) High entropy contexts: When the model is uncertain (i.e., the distribution is flat in temperature scaling), dropping one large token **frees up entropy space** and results in outputs that are more faithful to the overall model uncertainty rather than dominated by noise from a single strong guess.
>
> d) Filtering out “boring” or unhelpful dominant tokens: In local sampling, the single highest-probability token is often a structural or low-content token—punctuation, whitespace/“space” tokens, quotes, braces, or function words (“the”, “to”). These tokens are important for grammar, but they add little semantic variety. When high creativity is desired, letting one such token monopolize the shortlist crowds out several content-bearing alternatives. On the hand, dropping such a token can free up entropy budget for several **semantically rich alternatives** that drive the story, reasoning, or creativity forward.
>
> >**On the unnecessary complication of ECMM:**
>
> While the ECMM formulation may seem "unnecessary and complicated", we would emphasize on  our goal here: **to demonstrate the sampling problem's relevance as the entropy maximization problem, which is essential in understanding the theoretical underpinnings of sampling diversity and optimal token distribution**. Additionally, we would respectfully differ on the claim that "the complexity of the ECMM formulation would be unnecessary as the Greedy version is not the exact solution." There are two points to consider: firstly, the ECMM is a more accurate—albeit NP-hard—formulation that yields a principled perspective on the objective landscape and helps frame the underlying optimization problem rigorously, which we believe is essential in terms of understanding the problem in its core relevant format. Secondly, Greedy, on the other hand, acts as an efficient approximation to this formulation. **Neither of these perspectives has been previously explored in the literature in this context**. We thus strongly believe the problem and its proposed approximate and viable solutions merit the community's attention, particularly in light of our newly evaluated empirical results and theoretical insights. Nevertheless, we resonate with the reviewer’s broader philosophy, and would like to emphasize that, "**despite the theoretical establishment (which we believe is essential for the community to gain a comprehensive understanding), we also provide a practically viable approximate alternative that currently achieves state-of-the-art performance**." Demonstrating the Greedy approach only would have left the work under-motivated and lacking theoretical grounding, limiting its generalizability and long-term value for the community to investigate.

---

> > ### Comment · Reviewer_2JSm · 2025-08-07
> >
> > Regarding human evals:
> >
> > I will not pass judgment on the reviewers of min-p, but this does not change my concerns regarding the human evaluations (min-p had 54 participants). Human evals are expensive to run and would require more than a minor revision.
> >
> > Regarding the claim that ECMM is the right starting point:
> >
> > My take is: "you start with an overly complex setup and end up with a simple algorithm, which is then empirically evaluated". It is neither clear that the overly complex setup is necessary, given that there are other, more straightforward possible explanations, nor that it is indeed correct. Empirically, you'd have to show that a non-greedy version of ECMD/ECMM outperforms the greedy algorithm to show its merit. Given the high computational requirements, this seems hardly achievable/falsifiable, though.
> >
> > Overall, I'm remaining borderline reject. Thank you for your understanding.

---

> ### Author Response · Authors · 2025-08-05
> **Official comment to Reviewer 2JSm (part 2/2)**
>
> >**Clarification on Top-n = 100:**
>
> The use of top-n = 100 is an empirical preprocessing step aimed at improving computational efficiency by avoiding the processing of tokens with zero probability, which do not contribute to the entropy calculation. In all of our experiments, tokens ranked beyond the top 100 consistently have zero probability in the sorted token distribution. Therefore, truncating at this threshold allows us to safely discard these tokens without affecting the results. We would like to emphasize that this is purely an *implementation detail* and *does not impact* any of the theoretical arguments or conclusions presented in the paper.
>
> We would thus hope the reviewer to reassess their evaluations based on both the merit of the problem formulation followed by a practically viable solution, as we truly believe the draft's initial derivation is essential for the community to have a clear picture on the sampling problem, which has been largely missing in literature.
>
>
> Regards,
>
> Authors

---

> > ### Author Response · Authors · 2025-08-07
> > **Final note to reviewer 2JSm**
> >
> > Dear reviewer 2JSm,
> >
> > We remain thankful to your insightful reviewing comments and post rebuttal feedback. We do note that there may be some pertaining mis-understanding on why we attempted the problem with initial demonstration of criticality of the sampling problem, and associated theory. We thus did put our best and honest effort with necessary examples, evidences and justification around that. We also note that we have addressed your remaining doubts in the follow-up remarks. As the "author-reviewer discussion period" is ending in few hours, we hope you to provide your kind consideration to incorporate the substantial rebuttal evidences, that we believe has significantly strengthened the contributions of our state-of-the-art sampling method. We remain hopeful for the broader AI community to take this work forward as a new baseline to balance the *creativity and coherence of the generation outcomes*.
> >
> > Best
> >
> > Authors

---

### Official Review · Reviewer_ELnF · 2025-07-05

**Clarity:** 3
**Significance:** 3
**Originality:** 3
**Rating:** 5
**Confidence:** 4

**Summary:**

This paper introduces top-H decoding, a novel sampling strategy for LLMs that aims to balance creativity and coherence in text generation. The authors formulate this challenge as an entropy-constrained minimum divergence (ECMD) problem, which they prove equivalent to an entropy-constrained mass maximization (ECMM) problem. After showing ECMM is NP-hard, they propose a practical greedy algorithm that dynamically selects tokens based on entropy constraints. The method outperforms existing approaches, such as min-p and top-p sampling, achieving improvements in accuracy on creative writing benchmarks and maintaining robustness on reasoning tasks.

**Questions:**

1. **Theoretical justification for constraint form**: Why is the entropy constraint α·H(p) the most appropriate form? Have you considered alternative constraint formulations that might be more theoretically principled?

2. **Approximation guarantees**: Given the careful theoretical setup, can you provide any approximation bounds for the greedy algorithm, even in special cases? This would significantly strengthen the theoretical contribution.

3. **Computational analysis**: How does the computational overhead of top-H compare quantitatively to simpler methods like top-p and min-p? Can you provide timing comparisons and analysis of the trade-off between performance and efficiency?

4. **Human evaluation validation**: The heavy reliance on LLM-as-judge evaluation is concerning. Can you provide human evaluation studies to validate that the improvements measured by GPT-4o actually correspond to human preferences?

5. **Hyperparameter robustness**: How sensitive is performance to the choice of α across different models, tasks, and domains? Can you provide guidance for setting α beyond empirical grid search?

**Ethical Concerns:**

["NO or VERY MINOR ethics concerns only"]

**Final Justification:**

I think the paper is good for publication at NeurIPS despite these weaknesses, which were mentioned by Reviewer 2JSm. I don't find the overcomplicated theory a problem, but I agree that better human evaluation can make the paper stronger.

**Limitations:**

The authors acknowledge the limitation regarding lack of formal approximation guarantees in Appendix E. However, they could better address several other limitations: (1) the heavy reliance on automated evaluation without human validation, (2) the limited theoretical analysis of why the specific entropy constraint form is optimal, (3) incomplete analysis of computational overhead, and (4) potential sensitivity to hyperparameter choices across different domains.

**Quality:**

3

**Strengths And Weaknesses:**

**Strengths:**
1. **Strong theoretical foundation**: The paper provides a principled mathematical framework for the creativity-coherence tradeoff by formulating it as an optimization problem with entropy constraints.

2. **Comprehensive experimental evaluation**: Extensive experiments across multiple models (LLaMA3.1, Qwen2.5, Phi-3), benchmarks (Alpaca-Eval, MT-Bench, GSM8K, GPQA), and temperature settings demonstrate consistent improvements over baselines.

3. **Simple algorithm with strong empirical performance**: The greedy algorithm is simple to implement and shows robust performance, particularly at higher temperatures.

4. **Thorough analysis of design choices**: Good ablation studies on the α parameter and empirical validation that the greedy solution approximates the optimal solution.

5. **Temperature robustness**: Top-H maintains performance much better than alternatives as temperature increases, which is crucial for creative applications.

**Weaknesses:**
1. **Limited theoretical novelty**: While the formulation is sound, entropy-based sampling has been explored before (η-sampling). The main theoretical contribution is the specific ECMD formulation and NP-hardness proof, which are somewhat incremental.

2. **No approximation guarantees**: Despite the careful theoretical setup, the proposed greedy algorithm comes with no formal approximation bounds or worst-case performance guarantees, limiting the theoretical value.

3. **Questionable evaluation methodology**: Heavy reliance on LLM-as-judge evaluation without human validation raises concerns about evaluation validity. It can be good to make human evaluation similar to the min-p paper.

4. **Hyperparameter sensitivity**: The α parameter requires careful tuning, and while the paper shows α=0.4 works well empirically, there's limited theoretical guidance for choosing this value or adapting it across different contexts.

---

> ### Author Rebuttal · Authors · 2025-07-29
>
> We thank the reviewer  ELnF for the insightful feedback and clarify their concerns below.
>
> >**Justification for constraint form:**
>
> The choice of the entropy constraint form, $H(q) \le \alpha H(p)$, is central to our method's theoretical foundation and is chosen for its ability to dynamically balance creativity and coherence. We aim to develop a sampling framework explicitly aware of the model's confidence, not just based on the single highest-probability token, but on the overall shape of the probability distribution. Shannon entropy $H(p)$, is a measure for the uncertainty of the entire distribution $p$.
> Additionally, the formulation establishes an **adaptive upper bound on the randomness** of the selected token distribution, $q$. This way the generation can retain the following:
>
> *Adapts to Model Confidence:* The constraint is proportional to the entropy of the model's original prediction, instead of a fixed value. Thus, when the model is uncertain (the distribution is flat, with a high $H(p)$), it allows the sampling distribution $q$ to have a higher entropy,  encouraging  exploration and creativity when multiple tokens are plausible candidates. Alternately, when the model is confident (distribution with sharp peaks, with a low $H(p)$), it forces  the sampling distribution $q$ to have a lower entropy, preserving  coherence by restricting the sampling pool to high-probability tokens.
>
> *Provides Control:* The hyperparameter $\alpha \in$ (0,1) acts as a knob to control the trade-off between creativity and coherence.
>
> >**Exploration of alternative formulations:**
>
> We considered approaches including Mirostat [1],min-p [2], and top-p [3]. However, these approaches either fail to adapt to the model's confidence or lack a direct theoretical link to the distribution's overall uncertainty.
>
>
> >**Incremental contribution over $\eta$‑sampling**
>
> Despite both works control randomness via entropy, following are key differences:
>  $\eta$‑sampling specifies a *rule* (“clip until entropy < $\eta$”) without any optimization to balance between creativity and coherence while ECMD starts from first principles, *minimize divergence from the model distribution subject to an entropy cap*. We further demonstrate the divergence-mass equivalence for the first time with an information-theoretic justification.
> We analyze the *unrestricted subset selection* problem (no contiguity imposed on the order of token selection) and show it is NP-hard. This explains *why* a naive global optimizer is impractical motivating our greedy restriction. We further prove **monotone entropy growth** (Thm. 3) within probability-ordered prefixes giving a unique stopping point and justifying Top‑H. $\eta$‑sampling offers neither hardness results nor structural theorems.
> ECMD provide a **single objective** that subsumes and relates diverse decoding heuristics (T, nucleus, top‑k) under clear trade-off parameters $\alpha$. We quantify how close the greedy solution is to the true optimum (Fig. 5: conservative mean − $\sigma$ mass ratio ≥ 96%), while prior work lack these approximation results.
>
> >**Approximation guarantee:**
>
> Please refer to the **Discussion on formal approximation bound for the Top-H** response to reviewer bsFC on this.
>
> >**Computational overhead  and timing comparisons:**
>
> Table R1T1 provides the comparison of the average runtime per token (ms/token) of Top-H with Top-p and Min-p. We used LLaMA3.1-8B-Instruct, LLaMA3.3-70B-Instruct, and Phi-3-Mini-3.8B. For each setting, we averaged results over 100 prompts from the AlpacaEval, generating 128 tokens per prompt.
> ### R1T1
> |  **Temp** | **LLaMA-8B** | **LLaMA-8B**| **LLaMA-8B**| **Phi-3** | **Phi-3**  | **Phi-3**  | **LLaMA-70B** | **LLaMA-70B** |**LLaMA-70B**  |
> |:--:|:--:|:--:|:--:|:--:|:--:|:--:|:--:|:--:|:--:|
> |     | Top-H | Min-p | Top-p | Top-H | Min-p | Top-p | Top-H | Min-p | Top-p |
> | **1.0** | 28.3951 | 27.3396 | 27.4275 | 24.3847 | 23.6499 | 23.7809 | 219.3837 | 219.1391 | 218.4900 |
> | **2.0** | 28.4671 | 27.3840 | 27.4389 | 24.5929 | 23.9397 | 23.5844 | 219.3428 | 218.3609 | 217.7083 |
>
> Specifically, we observe a negligible overhead of as low as 0.8%.
>
> Complexity analysis: All methods begin with a sorting step, typically $\mathcal{O}(n ~ log ~ n)$.
> Top-p and Min-p performing CUMSUM and thresholding, respectively, both having a complexity of $\mathcal{O}(n)$. Below, we explain the incremental entropy accumulation of Top-H to be $\mathcal{O}(n)$.
> The Theorem 3 of the paper leads to the following algorithm for computing entropy incrementally:
> Let us define $\(h_{j-1} = \sum_{i=1}^{j-1} p_i \log p_i \)$, so the expression becomes:
> $H(q^{j-1}) = \log (\Gamma_{j-1}) - \frac{h_{j-1}}{\Gamma_{j-1}}$
> Now, incremental entropy accumulation pseudocode:
> ```
> Initialize Γ ← 0,  h ← 0,  H ← 0
> for each step j:
>     Γ ← Γ + p_j
>     h ← h + p_j log p_j
>     H ← log(Γ) - h / Γ
> ```
> This runs in $\mathcal{O}(n)$ time.
>
> >**Need for Human Eval:**
>
> We have now conducted Human Eval of LLM-generated texts using a setup similar to that of min-p, and compared with Top-p and Min-p. We recruited 14 PhD students for this. We used LLaMA3.1-8B-Instruct with texts generated using a prompt adapted from the min-p framework: “Write me a creative story.”  For each configuration, we generated three outputs, capped at 512 tokens. Participants were asked to evaluate them along quality and diversity with rating on a scale of 1-10. Results are shown in R1T2.
>
> ### R1T2
> | Sampling Method | Creativity (T = 0.7) | Coherence (T = 0.7) | Creativity (T = 1.0) | Coherence (T = 1.0) | Creativity (T = 2.0) | Coherence (T = 2.0) |
> |:--:|:--:|:--:|:--:|:--:|:--:|:--:|
> | Top-p        |      7.57 $\pm$ 0.72               |    4.78 $\pm$ 0.93                |         6.28 $\pm$ 0.69             |     5.78 $\pm$ 0.79                |        3.57 $\pm$ 0.72              |         7.57 $\pm$ 0.62            |
> | Min-p        |      6.92 $\pm$ 0.59                |    5.28 $\pm$ 1.03                 |       6.21 $\pm$ 0.77               |      5.92 $\pm$ 0.79               |     6.00 $\pm$ 0.75                 |   6.57 $\pm$ 0.72                  |
> | Top-H        |       7.35 $\pm$ 0.71               |      5.42 $\pm$ 0.62               |      6.57 $\pm$ 0.90                |         6.42 $\pm$ 0.91            |       6.42 $\pm$ 0.62               |   7.07 $\pm$ 0.70                  |
>
> > **Hyperparameter robustness:**
>
> Please note that Figure 4 of the draft shows the creativity-coherence trade-off across a range of $\alpha$ values. Specifically, the changes in coherence and creativity metrics are gradual with changes in $\alpha$, indicating it to not be overly sensitive to small adjustments. Intuitively, $\alpha$ controls the balance between confidence and diversity in the decoding, with small $\alpha$ encouraging more deterministic outputs while its large value increasing the randomness and thus encouraging creativity.
> This controllability gives users practical guidance: moderate $\alpha$ values (around 0.4) offer a balanced trade-off, and users can adapt $\alpha$ based on their preference for coherence versus creativity.
> Additionally, the following table R1T3 reports GPQA accuracy and the average sampling pool size across different $\alpha$ values. The experiment is done using LLaMA3.1-8B-Instruct model with temperature T = 1.5. These results show that: 1. Larger $\alpha$ values slightly reduce accuracy, which aligns with the nature of GPQA’s graduate-level questions that benefit from more confident (less diverse) answers. 2. Sampling pool size increases with $\alpha$, providing more generative options and supporting the creativity aspect observed in Figure 4.
> ### R1T3
> |$~~~~~~~~~~~~~~~~~~~~~~~\alpha$|0.10|0.15|0.20|0.25|0.30|0.35|0.40|0.45|0.50|0.55|0.60|0.65|0.70|0.75|0.80|0.85|0.90|
> |:-:|:-:|:-:|:-:|:-:|:-:|:-:|:-:|:-:|:-:|:-:|:-:|:-:|:-:|:-:|:-:|:-:|:-:|
> |Accuracy on GPQA(%)|0.3085|0.3085|0.3095|0.3085|0.3125|0.3103|0.3058|0.305|0.2869| 0.2937|0.2879|0.279|0.2655|0.2879|0.2612|0.2656|0.2701|
> |Average sampling pool size|1.01|1.03|1.20|1.48|1.90|2.53|3.48|4.74|6.94|9.11|11.79|15.77|21.35|27.99|36.28 |47.06|59.28|
>
> Importantly, both accuracy and sampling pool size show smooth and consistent trends, reinforcing the method’s robust behavior. For moderate $\alpha$, accuracy changes smoothly; the small fluctuations at higher $\alpha$ are expected, as higher $\alpha$ increases the sampling pool and thus randomness.
>
> ---
> ## References
>
> [1] Mirostat: A neural text decoding algorithm that directly controls perplexity, arXiv.
>
> [2] Turning up the heat: Min-p sampling for creative and coherent llm outputs, arXiv .
>
> [3] The curious case of neural text degeneration, arXiv .

---

> > ### Comment · Reviewer_ELnF · 2025-08-07
> >
> > Hi,
> > Thank you for the detailed response and clarification. You addressed my concerns.

---

> ### Author Response · Authors · 2025-08-04
>
> Dear Reviewer,
>
> Thank you for the time and effort you’ve put into reviewing our submission. We’ve carefully considered each of your comments and concerns, and have done our best to address them thoroughly in our rebuttal with the necessary theoretical or empirical evidences.
>
> We understand that time-bound reviewing process is demanding. However, as the deadline of discussion is approaching soon, we would greatly appreciate it if you could take a moment to revisit our response. With the enhanced thoroughness driven by your feedback, we believe top-H decoding has emerged as a promising alternative to the existing approaches.
>
> Your feedback is invaluable and can help ensure that our clarifications are appropriately considered in the final decision.
>
> Please don’t hesitate to reach us out in case any further discussion is needed.
>
> Best regards,
>
> Authors

---

> ### Author Response · Authors · 2025-08-07
> **Final reminder request**
>
> Dear reviewer ELnF,
>
> We remain thankful to your insightful reviewing comments. We provided our best attempt to address your remaining concerns with necessary evidences in the rebuttal. As the "author-reviewer discussion period" is ending in few hours, we hope you to provide your kind consideration to incorporate the substantial rebuttal evidences, that we believe has significantly strengthened the contributions.
>
> Best
>
> Authors

---

> > ### Comment · Area_Chair_9X9K · 2025-08-07
> > **Please Engage in Author Response Discussion**
> >
> > Hi Reviewer ELnF,
> >
> > We strongly encourage you to review the authors’ rebuttals and see how they’ve addressed your comments.
> > Your participation helps ensure a fair and thoughtful review process.
> >
> > Best,
> > AC

---

### Note · Authors · 2025-08-15

We sincerely thank the PC, SAC, AC, and reviewers for their efforts in running the reviewing process. During the rebuttal, we made substantial efforts to address the reviewers’ concerns, and are glad to see we could address the majority of the concerns of all the reviewers. We understand to have an acceptance score from two reviewers(reviewer **ELnF** and **bsFC**), while there might be some general misunderstanding with the reviewer **2JSm** who gave a borderline rejection. Though the reviewers acknowledged that we could address their concerns irrespectively. We summarize our rebuttal response here:

 We provided additional experimental results on **Human Evaluation** (compared with the recent most ICLR'25 ORAL paper to demonstrate top-H's superiority), **Validation of top-H with a larger (70B) model**, **Computational cost of top-H**, **Statistical significance of the results**, and the **justification of the hyperparameter $\alpha$**. We also presented a comprehensive theoretical analysis of the **worst-case performance bound** of top-H relative to the optimal solution, thereby strengthening its theoretical foundation.
 Furthermore, we offered explicit justification for the **rationale behind the ECMD and ECMM formulations** and identified concrete scenarios where the optimal solution is desirable. We are grateful that the reviewers found our theoretical framework and empirical results compelling, and that their concerns have been satisfactorily addressed.

We remained transparent in our problem formulation, even though we understood that a simpler approximation could have been a decent alternative, because we believe it is important for the broader AI community to have a clear understanding of the problem, regardless of the practical deployability of the proposed solution. We believe the reviewers and AC to be inline with our transparent approach of problem formulation. Thus in summary, given the strong theoretical motivation for top-H and its substantial empirical gains compared to SoTA, we believe this work deserves the right attention of the AI broader community and inspire further research on creative/coherent text generation.

---

### Decision · Program_Chairs · 2025-09-17

**Decision:**

Accept (poster)

**Comment:**

This paper proposes a new sampling strategy for LLMs, named top-H decoding, which aims to balance creativity and coherence in text generation. The authors prove that the underlying problem is NP-hard and derive a greedy algorithm as a practical solution.

The main concerns raised by the reviewers, including the lack of approximation guarantees, absence of human evaluation, and limited analysis on hyper-parameter sensitivity, were addressed in detail in the author rebuttal.

Reviewer opinions diverged significantly: two reviewers were satisfied with the rebuttal and expressed support for acceptance, while one reviewer maintained strong reservations, arguing that the method should have been described in a simpler way. The authors acknowledged that a simpler presentation is possible but emphasized the importance of providing the full theoretical background, a position with which the area chair agrees.

Taking these points into account, the area chair recommends acceptance of the submission. The additional experimental results and clarifications provided in the rebuttal should be incorporated into the final version.